# Diabetes downregulates the antimicrobial peptide psoriasin and increases *E. coli* burden in the urinary bladder

Soumitra Mohanty [1,2], Witchuda Kamolvit[1,2], Andrea Scheffschick[3], Anneli Björklund[4,5], Jonas Tovi[6], Alexander Espinosa[3], Kerstin Brismar[5], Thomas Nyström[7], Jens M. Schröder [8], Claes-Göran Östenson [5], Pontus Aspenström[9], Hanna Brauner[3,10] & Annelie Brauner [1,2] ✉

Diabetes is known to increase susceptibility to infections, partly due to impaired granulocyte function and changes in the innate immunity. Here, we investigate the effect of diabetes, and high glucose on the expression of the antimicrobial peptide, psoriasin and the putative consequences for *E. coli* urinary tract infection. Blood, urine, and urine exfoliated cells from patients are studied. The influence of glucose and insulin is examined during hyperglycemic clamps in individuals with prediabetes and in euglycemic hyperinsulinemic clamped patients with type 1 diabetes. Important findings are confirmed in vivo in type 2 diabetic mice and verified in human uroepithelial cell lines. High glucose concentrations induce lower psoriasin levels and impair epithelial barrier function together with altering cell membrane proteins and cytoskeletal elements, resulting in increasing bacterial burden. Estradiol treatment restores the cellular function with increasing psoriasin and bacterial killing in uroepithelial cells, confirming its importance during urinary tract infection in hyperglycemia. In conclusion, our findings present the effects and underlying mechanisms of high glucose compromising innate immunity.

The high prevalence of diabetes is a major global health challenge[1], often accompanied with increased risk of bacterial infections[2]. In particular, *E. coli* urinary tract infections (UTI) are common and more frequently associated with serious complications, such as urosepticemia[3]. Traditionally, glycosuria is believed to provide an optimal environmental condition for bacterial growth[4]. However, several factors like age, poor metabolic control, long term complications like neuropathy with incomplete bladder emptying as well as diabetic nephropathy contribute to the risk for UTI[5,6]. Moreover,

immunogenic impairment like reduced migration and chemotaxis of leucocytes play a role in the severe UTI pathogenesis of diabetic patients.

The host cells are well equipped with defense mechanisms shielding against invading microorganisms. In recent time, several factors have been identified that can protect the bladder from invading pathogens. Antimicrobial peptides (AMPs), part of first line innate immune response, defend the urothelium from pathogens and have the potential to become new UTI therapies[7,8]. They are expressed by

[1]Department of Microbiology, Tumor and Cell Biology, Karolinska Institutet, Stockholm, Sweden. [2]Division of Clinical Microbiology, Karolinska University Hospital, Stockholm, Sweden. [3]Department of Medicine, Solna, Stockholm, Sweden. [4]Center for Diabetes, Academic Specialist Center, Stockholm County Council, Solna, Sweden. [5]Department of Molecular Medicine and Surgery, Karolinska Institutet, Stockholm, Sweden. [6]Capio Health Care Center, Solna, Sweden. [7]Department of Clinical Science and Education, Division of Internal Medicine, Unit for Diabetes Research, Karolinska Institutet, South Hospital, Stockholm, Sweden. [8]Department of Dermatology, Venerology and Allergology, University Hospital Schleswig-Holstein, Kiel, Germany. [9]Rudbeck Laboratory, Department of Immunology, Genetics and Pathology (IGP), Uppsala University, Uppsala, Sweden. [10]Dermato-Venereology Clinic, Karolinska University Hospital, Stockholm, Sweden. ✉e-mail: Annelie.Brauner@ki.se

epithelial and endothelial cells[9] as well as immune cells like neutrophils[10]. We and others have previously reported the impact of human antimicrobial peptides cathelicidin, hBD1, hBD2 and RNase7 in the urinary tract[8,11–13]. Most of these AMPs are cationic in nature and active against a wide range of both Gram-positive and Gram-negative bacteria. The antimicrobial peptide psoriasin, encoded by *S100A7* is a

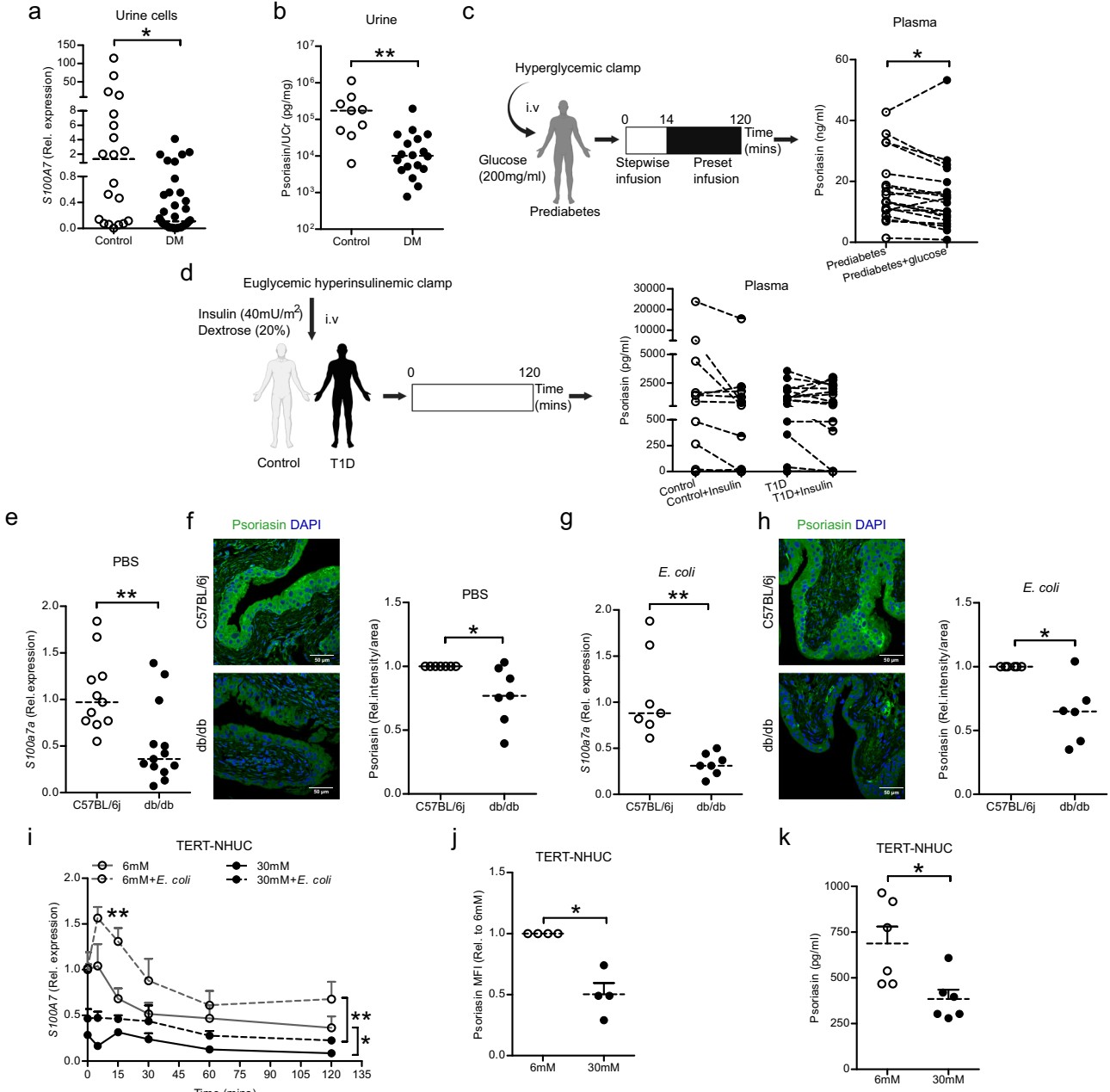

**Fig. 1 | High glucose downregulates expression of psoriasin in plasma and urinary bladder. a** Expression of *S100A7* mRNA in urine cells of patients with diabetes (DM), ($n = 36$) and non-diabetic individuals ($n = 20$) (unpaired two-tailed t test, $p = 0.0164$). **b** Urine psoriasin normalized to urine creatinine (Ucr), in DM ($n = 18$) and non-diabetic individuals ($n = 9$) (unpaired two-tailed t test, $p = 0.0094$). **c** Schematic presentation of hyperglycemic clamps in prediabetes ($n = 19$), plasma psoriasin analysis before and 2 h after i.v. glucose infusion (paired two-tailed t test, $p = 0.0264$). **d** Schematic presentation of euglycemic, 5 mM, hyperinsulinemic clamp in type 1 diabetes (T1D) ($n = 23$) and healthy ($n = 18$) individuals, plasma psoriasin analysis before and after insulin treatment (paired two-tailed t test). Expression of **e** *S100a7a* mRNA (unpaired two-tailed t test, $p = 0.0044$) and **f** protein in PBS treated diabetic, db/db ($n = 13$; $n = 7$) and non-diabetic, C57BL/6j ($n = 11$; $n = 7$) mice respectively (Mann-Whitney two-tailed test, $p = 0.0204$). Expression of **g** *S100a7a* mRNA (unpaired two-tailed t test, $p = 0.0017$) and **h** protein in 24 h *E. coli* infected, db/db ($n = 7$; $n = 6$) and C57BL/6j ($n = 7$; $n = 6$) respectively (Mann-Whitney two-tailed test, $p = 0.0493$). **i** *S100A7* mRNA 5 ($n = 6$

infected; $n = 8$ uninfected), 15 ($n = 6$ infected; $n = 6$ uninfected), 30 ($n = 6$ infected; $n = 6$ uninfected), 60 ($n = 6$ infected; $n = 6$ uninfected) and 120 mins ($n = 6$ infected; $n = 6$ uninfected) post *E. coli* infection at MOI 10 or medium only after 24 h glucose treatment (normal = 6 mM: high=30 mM) of TERT-NHUC uroepithelial cells, compared to time 0 ($n = 2$), mean value is presented (unpaired two-tailed t test, $p = 0.0072$ and One-way ANOVA, multiple comparison test, $p \leq 0.05$, $p \leq 0.01$ respectively. **j** Intracellular psoriasin levels (mean fluorescence intensity, MFI, flowcytometry) after 36 h glucose treatment of TERT-NHUC, ($n = 4$) (Mann-Whitney two-tailed test, $p = 0.0202$). **k** Secretion of psoriasin after 24 h treatment with normal and high glucose of TERT-NHUC cells ($n = 6$) (unpaired two-tailed t test, $p = 0.0165$). In vitro experiments were performed in duplicate or triplicate with at least 3 independent experiments, presented as mean ± SEM, statistical outliers defined by Grubb's test were excluded. For in vivo and human material analysis, individual values and median are shown, *$p < 0.05$ and **$p < 0.01$. Source data are provided as a source data file.

member of the S100 protein family and has been detected in the urinary tract[14]. It is mainly known for its high antibacterial activity against *E. coli*, sequestering zinc which restricts the bacterial growth[15].

Although the importance of psoriasin during *E. coli* infections is recognized, the possible activity in diabetes and during high glucose is not yet known. We here sought to investigate the impact of glucose on psoriasin and the pathogenesis of *E. coli* UTI, with emphasis on the uroepithelium and defense strategies in the urinary bladder during diabetes.

## Results

### High glucose decreases psoriasin in the serum/plasma and urinary bladder

The possible impact of high glucose on antimicrobial peptides was analyzed in human uroepithelial cells TERT-NHUC and resulted in significantly lower expression of *S100A7, DEFB4A* and *RNASE7*, while *CAMP, DEFB1* and *DEFB103A* remained unchanged compared to low glucose (Fig S1a). Since *DEFB4A*[16] and *RNASE7*[17] have previously been demonstrated to be compromised by high glucose, we focused on *S100A7*, psoriasin. To ensure the clinical relevance, we investigated the psoriasin levels in patients with diabetes (Table S1). In these patients, we observed downregulation of *S100A7* mRNA in urine exfoliated cells (Fig. 1a) as well as lower psoriasin protein levels in urine (Fig. 1b) with a similar trend of serum psoriasin levels compared to non-diabetic controls (Fig. S1b).

To investigate the possible impact of glucose and insulin on psoriasin, hyperglycemic[18] and euglycemic hyperinsulinemic clamps were used[19]. Hyperglycemic clamps were performed in prediabetic individuals with a high risk of developing type 2 diabetes (T2D). In these clamp studies, glucose infusion for 2 h resulted in increased median plasma glucose from 5.4 (range 4.6–6.4) mM to 11.4 (range 10.8–12.5) mM. This stimulated insulin secretion, augmenting median plasma insulin levels from 16.1 (range 8.4-74.2) mU/L to 95.2 (range 35.6–160.0) mU/L[18]. These changes were accompanied with decreased psoriasin levels (Fig. 1c). Contrary, euglycemic hyperinsulinemic clamps in patients with type 1 diabetes (T1D) and in non-diabetic individuals[19], showed no difference in the plasma psoriasin levels (Fig. 1d). Hence, the lower plasma psoriasin was associated with enhanced blood-glucose levels, but not with insulin levels.

To confirm our results in vivo, female db/db mice with T2D and C57BL/6j non-diabetic control mice were studied. The *S100a7a* mRNA (Fig. 1e) and psoriasin protein levels (Fig. 1f), were significantly lower in diabetic mouse bladders, with the peptide primarily localized in the superficial umbrella cell layers. To further investigate the effect on psoriasin during infection in diabetes, mice were transurethrally infected with *E. coli*. We observed lower expression of *S100a7a* at the mRNA (Fig. 1g), and protein levels (Fig. 1h) at 24 h and 7 days post infection (Fig. S1c, d) in urinary bladders of diabetic compared to control mice.

Similarly, TERT-NHUC uroepithelial cells cultured under normal glucose condition showed a rapid increase of *S100A7* mRNA peaking already after 15 min of *E. coli* infection, (Fig. 1i). On the other hand, TERT-NHUC cells cultured in high glucose expressed lower *S100A7* mRNA (Figs. 1i and S1e), with similar results in 5637 uroepithelial cells (Fig. S1f). In line with our mRNA data, lower cytoplasmic and vesicle associated psoriasin (Figs. 1j and S1g) as well as secreted psoriasin (Fig. 1k) protein were also observed in high glucose treated TERT-NHUC cells.

### High glucose impairs IL-6 mediated psoriasin expression

Since cytokines are known to regulate antimicrobial peptides[20], we hypothesized that not only psoriasin, but also cytokines could be compromised by high glucose. In line with this, the proinflammatory cytokines, *IL1B* (Fig. 2a) and *IL6* (Fig. 2b) were downregulated on the mRNA level in urine exfoliated cells from patients with diabetes. Moreover, the protein levels of IL-1β and IL-6 were lower and located in

the superficial umbrella cells (Fig. 2c) in diabetic mice bladders 24 h post *E. coli* infection. Likewise, in high glucose treated TERT-NHUC uroepithelial cells, the expression of *IL1B* and *IL6* mRNA (Fig. 2d, e) and IL-6 protein (Fig. 2f) was compromised. Although, IL-6 is a known regulator of pSTAT3/STAT3, no difference of pSTAT3 was observed between high and low glucose treated TERT-NHUC cells substituted with 50 ng/ml of human IL-6 peptide (Fig. 2g). In high glucose treated cells, the expression of *SOCS3* mRNA, down-stream of STAT3, was compromised (Fig. 2h) possibly due to the increased expression of Aryl hydrocarbon receptor (*AHR*) mRNA in 5637 cells (Fig. 2i), which is known to downregulate SOCS3[21] and also to be upregulated by high glucose[22]. Interestingly, *SOCS3* mRNA was upregulated by IL-6 peptide treatment (Fig. 2j) in TERT-NHUC cells. Further to confirm the effect of IL-1β on IL-6 expression, supplementation with 20 ng/ml IL-1β peptide resulted in increased expression of both *IL6* and *S100A7* mRNA, which was quenched by diacerein, a specific IL-1β blocker (Fig. 2k, l). IL-6 and psoriasin expressions have been shown to depend on each other in various cells[23,24]. We therefore speculated that high glucose would lead to lower IL-6 levels which could affect psoriasin expression. Therefore, high glucose treated TERT-NHUC cells were treated with recombinant IL-6 peptide. We demonstrated that psoriasin was upregulated on both mRNA (Fig. 2m) and protein levels (Fig. 2n), confirming the inter-relationship between IL-6 and psoriasin also during hyperglycemia.

### High glucose compromises occludin expression in plasma and urinary bladder

Antimicrobial peptides have been shown to regulate epithelial barrier function[25]. We therefore reasoned that the lower psoriasin levels may impact the expression of the tight junction protein, occludin. In line with this hypothesis, urine exfoliated cells from patients with diabetes, demonstrating lower levels of psoriasin, presented a clear down-regulation of *OCLN* (Fig. 3a). Likewise, plasma from hyperglycemic clamped individuals with prediabetes, showed decreased occludin levels (Fig. 3b).

Similarly, we observed lower *Ocln* on the mRNA (Fig. 3c) and protein levels in the diabetic *vs* non-diabetic mice, with the protein localized in the upper superficial umbrella cell layers of the bladder (Fig. 3d), 24 h (Fig. 3e, f) and 7 days (Fig. S2a, b) post *E. coli* infection. Our results were further confirmed in vitro, where uroepithelial cells, TERT-NHUC, exposed to high glucose down regulated occludin (Fig. 3g) at the mRNA and protein levels (Fig. 3h). To confirm the role of psoriasin in occludin expression, *S100A7* was deleted in TERT-NHUC cells using the crispr/cas9 system which resulted in lower expression of occludin (Fig. 3i) without inducing any adverse effect on cells as evident from nuclear and cytoskeleton integrity (Fig. S2c). Further-more, to verify the effect of psoriasin, TERT-NHUC uroepithelial cells treated with high glucose and supplemented with additional psoriasin peptide showed increased expression of *OCLN* mRNA (Fig. 3j). The Cys reduced form of psoriasin is a powerful endogenous zinc-chelator[26]. We therefore speculated that psoriasin could have a regulatory function in zinc homeostasis and its zinc-binding properties would impact the occludin gene expression[27]. To test if the zinc-binding property of psoriasin could explain the observed effects, the cell penetrating and zinc specific chelator, N,N,N',N'-tetrakis (2-pyridylmethyl) ethylene-diamine (TPEN) was used. Human uroepithelial cells, TERT-NHUC were treated with TPEN, followed by high glucose for a total 24 h. TPEN treatment resulted in increased *OCLN* mRNA (Fig. 3k) similar to the effect of the psoriasin peptide (Fig. 3j) indicating a possible role of intracellular zinc depletion in the increased expression of *OCLN* in high glucose treated human uroepithelial cells.

### High glucose and *E. coli* infection modulate membrane proteins

Since TERT-NHUC cells exposed to high glucose did not mount a pronounced psoriasin response when infected with *E coli*, we specu-lated that also other factors involved in bacterial infection could be

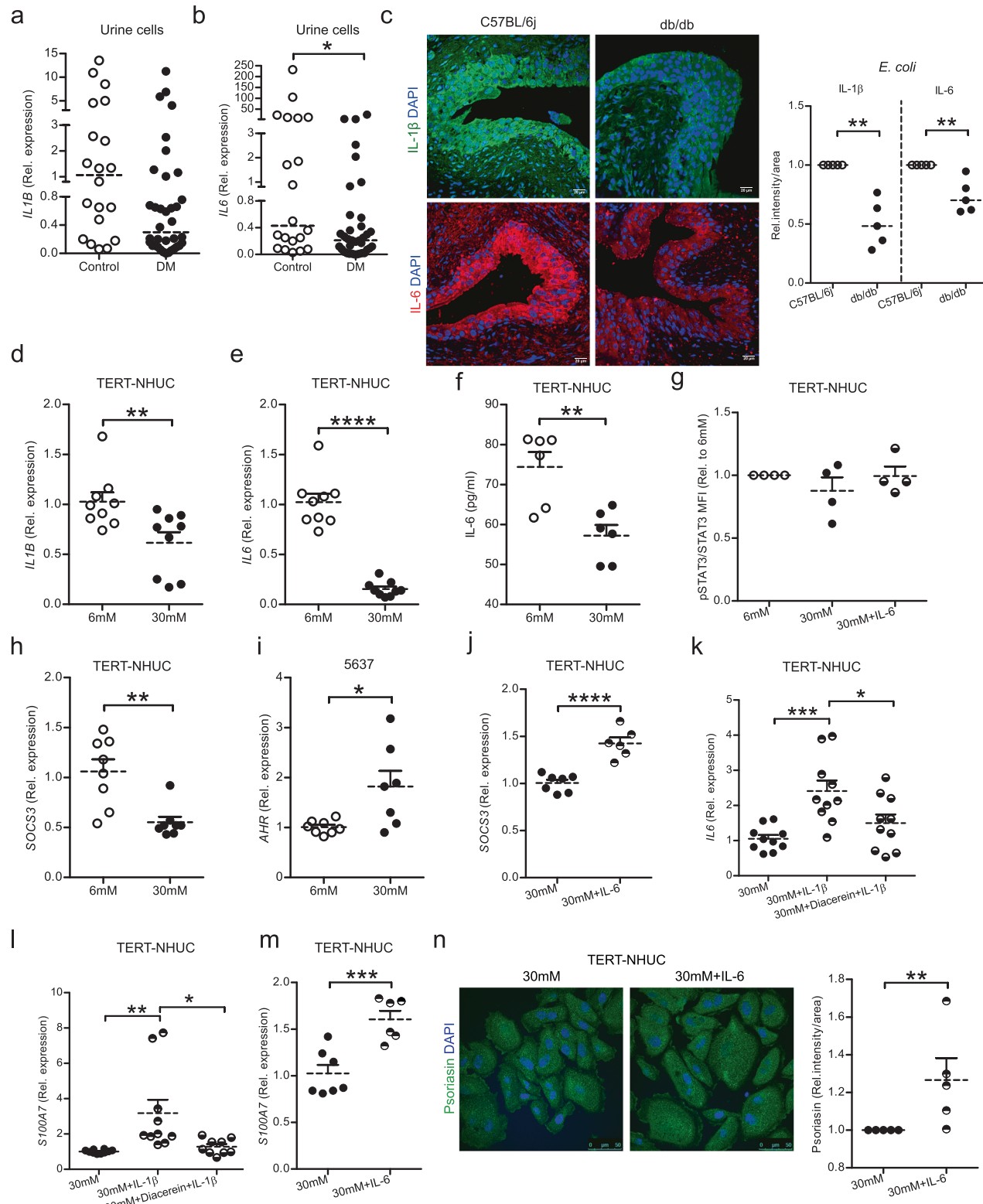

affected. Mannose or mannose-like receptors (MRC1) are known to play an important role in bacterial attachment to uroepithelial cells[28]. In the uninfected state, expression of *Mrc1* were similar in bladders of both types of mice. After 24 h of *E. coli* infection, however, diabetic but not non-diabetic control mice showed significantly higher expression of *Mrc1* both on the mRNA (Fig. S2d) and protein levels (Fig. 4a). In line with these results, TERT-NHUC uroepithelial cells treated with high glucose and infected with *E. coli* showed increased expression of *MRC1*

at the mRNA (Fig. S2e) and protein levels (Fig. 4b). After bacterial attachment to uroepithelial cells, caveolin 1 influences the endocytic uptake of *E. coli*, by forming a flask shaped caveolae[29]. Correspondingly, diabetic mice bladders showed clear upregulation of caveolin 1 (Fig. 4c), with similar result in high glucose treated TERT-NHUC uroepithelial cells (Fig. 4d). Based on our findings, we hypothesized that not only glucose, but possibly also psoriasin could impact membrane proteins. However, psoriasin peptide had no effect on *MRC1* in TERT-

**Fig. 2 | Effect of high glucose on cytokines and psoriasin expression.** Expression of **a** *IL1B* (unpaired two-tailed t test,) and **b** *IL6* mRNA in urine cells of patients with diabetes (DM), (*n* = 36) and non-diabetic individuals (*n* = 20) (unpaired two-tailed t test, *p* = 0.0465). Expression of **c** IL-1β and IL-6 (Mann-Whitney two-tailed test, *p* = 0.0075) expression in urinary bladders 24 h post *E. coli* infected diabetic, db/db (*n* = 5), and non-diabetic, C57BL/6j mice (*n* = 5). **d** *IL1B* and **e** *IL6* (*n* = 9) (unpaired two-tailed t test, *p* = 0.0100, *p* ≤ 0.0001 respectively) mRNA level in TERT-NHUC uro-epithelial cells cultured with normal and high glucose (normal = 6 mM: high = 30 mM) for 24 h. Secretion of **f** IL-6 measured from TERT-NHUC after 24 h glucose treatment (*n* = 6) (unpaired two-tailed t test, *p* = 0.0036). **g** Flow cytometric analysis of pSTAT-3 and total STAT3 (mean fluorescence intensity, MFI) after 36 h of glucose treatment and 1 h of 50 ng/ml of IL-6 peptide (*n* = 4) (One-way ANOVA, multiple comparison). **h** *SOCS3* (*n* = 8), and **i** *AHR* (6 mM, *n* = 8; 30 mM, *n* = 7) mRNA level in TERT-NHUC and 5637 cultured with normal and high glucose for 24 h (unpaired two-tailed t test, *p* = 0.0019, *p* = 0.0161), respectively. **j** Expression of *SOCS3* mRNA in high glucose

and IL-6 peptide (50 ng/ml) treated TERT-NHUC cells after 24 h (unpaired two-tailed t test, *p* ≤ 0.0001), (30 mM, *n* = 7; 30 mM+IL-6, *n* = 6). Expression of **k** *IL6* (*n* = 10) and **l** *S100A7* (30 mM and 30 mM+IL-1β, *n* = 10; 30 mM+Diacerein+IL-1β, *n* = 9) mRNA in high glucose and IL-1β peptide (20 ng/ml), in diacerein (50 μM) pretreated TERT-NHUC cells after 24 h (One-way ANOVA, multiple comparison, *p* ≤ 0.05, *p* ≤ 0.01 and *p* ≤ 0.001 respectively). **m** Expression of *S100A7* mRNA (30 mM, *n* = 7; 30 mM+IL-6, *n* = 6) (unpaired two-tailed t test, *p* ≤ 0.001), **n** and psoriasin protein levels in high glucose and IL-6 peptide (50 ng/ml) treated TERT-NHUC cells (Mann-Whitney two-tailed test, *p* = 0.0075) after 24 h and 36 h respectively (*n* = 5). In vitro experiments were performed in either duplicate or triplicate with at least 3 independent experiments, presented as mean ± SEM, statistical outliers defined by Grubb's test were excluded. For in vivo and human material analysis, individual values and median is mentioned, *\*p* < 0.05, *\*\*p* < 0.01, *\*\*\*p* < 0.001 and *\*\*\*\*p* < 0.001. Source data are provided as a source data file.

NHUC cells (Fig. S2f), while substitution reversed the hyperglycemia induced upregulation of *CAV1* (Fig. 4e), suggesting that by adding psoriasin the effect of high glucose could be outcompeted.

## High glucose reorganizes the cytoskeleton and translocate YAP/TAZ into the nucleus

Rho GTPases and caveolin 1 are known to mediate actin cytoskeletal rearrangements[30]. Rho proteins can be regulated by localizing to caveolae and interacting with caveolins[31]. Given the effect on caveolin 1, we speculated that high glucose and psoriasin might also influence RhoB and the cytoskeleton. In urine exfoliated cells from patients with diabetes, we observed decreased expression of *RHOB* mRNA (Fig. 5a). Furthermore, we confirmed decreased expression of *RHOB* mRNA (Fig. S3a) and protein (Fig. 5b) in high glucose exposed human uroepithelial cells, 5637. However, psoriasin peptide did not induce a direct effect on *RHOB* mRNA expression in high glucose treated cells (Fig. S3b). Interestingly, high glucose treatment of human uroepithelial 5637 cells revealed less long stress fibers, more cortical actin and predominantly nuclear YAP/TAZ (Fig. 5c–e), a transcription factor responsible for downregulation of psoriasin expression[32]. Contrary, in normoglycemic cells, YAP/TAZ was localized in both cytoplasm and nucleus with normal cell shape and long stress fibers. The same observation was found in TERT-NHUC uroepithelial cells (Fig. S3c–e). To confirm the role of RhoB on YAP/TAZ translocation, dominant negative RHOB/T19N and control RHOA/T19N constructs were separately transfected in high glucose treated 5637 cells followed by validation of nuclear YAP/TAZ. RHOB/T19N expression resulted in increased nuclear YAP/TAZ, whereas no effect with RHOA/T19N was observed, confirming a role of dominant negative RhoB in increasing YAP/TAZ translocation to the nucleus (Fig. 5f) with no effect on actin cytoskeleton (Fig. S3f). Moreover, the effect of IL-6 in TERT-NHUC cells cultured with high glucose resulted in YAP/TAZ translocation with less nuclear YAP/TAZ (Fig. 5g).

The effect of RhoB on intracellular bacterial communities (IBCs) was investigated in *E. coli* infected mice bladders. The significant reduction of Rhob was associated with higher degree of IBCs in diabetic mice bladder when compared to nondiabetic mice bladder (Fig. S3g).

## Effect of estradiol on high glucose induced bacterial load and clearance

To investigate the impact of glucose on infection and bacterial clearance, diabetic and non-diabetic control mice were infected with *E. coli*. Higher bacterial load was observed in the urine (Fig. 6a) as well as in the bladder (Fig. 6b) of diabetic mice, 24 h post-infection. Superficial bladder epithelial cells of all diabetic mice had numerous IBCs partly occupying the epithelial surface and destroying the cellular lining whereas, non-diabetic control mice had intact cellular lining with rare IBCs (Fig. 6c).

One and even two weeks after onset of infection, diabetic mice still had higher bacterial load in the urine (Fig. S4a and Fig. 6d) and ≥10⁶ bacteria in the bladder (Fig. S4b and Fig. 6e) with numerous IBCs and fused bacterial inclusions occupying the epithelium (Fig. S4c, d), indicating impaired bacterial clearance. To confirm the relevance of psoriasin activity in urine; samples from controls and patients with T2D were supplemented with 5 μM of psoriasin peptide. Significant *E. coli* killing was observed in both control and T2D urine samples (Fig. 6f). Further to confirm psoriasin mediated *E. coli* killing, blocking of psoriasin with a specific monoclonal antibody resulted in 2-fold higher bacterial numbers in normal glucose lysate. This was not observed in high glucose, which partly could be attributed to the initial low psoriasin levels (Fig. 6g).

To elucidate the impact of glucose in vitro, uroepithelial cells, 5637 (Fig. 6h) and TERT-NHUC (Fig. S4e) treated with high glucose were infected with *E. coli* showed increased bacterial adhesion and intracellular bacterial load compared to normal glucose concentrations.

It is worthy of note that estradiol did not directly influence the *IL1B* mRNA (Fig. 6i), while the expression of *IL6* mRNA (Fig. 6j) was increased, indicating that estradiol targets IL-6, followed by elevated *S100A7* on the mRNA (Fig. 6k) and protein levels (Fig. 6l). Further, increased bacterial killing was observed, compared to 5637 uroepithelial cells exposed to high glucose alone (Fig. 6m). Estradiol treatment of uroepithelial 5637 cells exposed to high glucose, inhibited long stress fiber formation and short bundles and increased cortical actin formation after 48 h (Fig. 5c, d). Moreover, estradiol rescued high glucose treated cells with YAP/TAZ being localized in both the cytoplasm and nucleus (Fig. 5e). Though estradiol further increased the amount of cortical actin, treatment still resulted in less bacterial load (Fig. 6m). To verify the effect of the actin network on the bacterial load, the actin inhibitor Latrunculin B was used, resulting in significantly reduced bacterial load, similar to estradiol treated cells (Fig. S4f). This further confirmed that in spite of the actin rearrangement the effect on the bacterial load was influenced by the bactericidal effect of the estradiol induced psoriasin.

## Discussion

We here demonstrate that high glucose levels compromised the innate immune response and impaired epithelial integrity. These findings may offer an explanation of the clinical observation that patients with poorly controlled diabetes have higher risk of recurrent UTI, acute pyelonephritis and urosepticemia[3]. Thus, confirming that impaired glycemic control may contribute to enhanced risk of infections[33].

Our current results demonstrate that high glucose inhibits plasma psoriasin in prediabetic individuals during hyperglycemic clamp. Lower serum levels of AMPs have however previously been observed in diabetic patients[16,17,34]. While insulin is known to regulate the

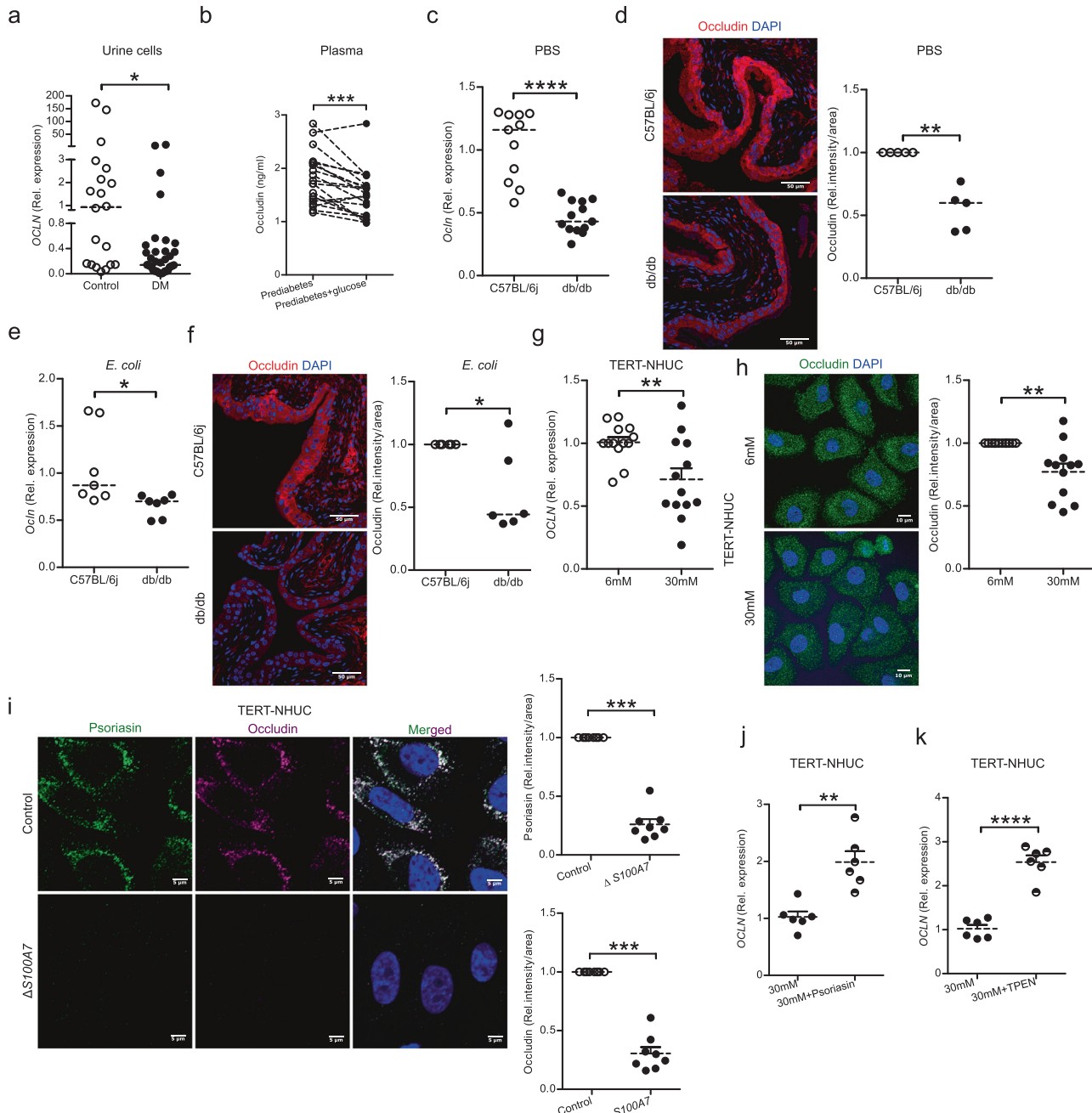

**Fig. 3 | Effect of diabetes and high glucose on occludin. a** Expression of *OCLN* mRNA in urine cells from non-diabetic (*n* = 20) and diabetic patients, DM (*n* = 36) (unpaired two-tailed t test, *p* = 0.0391). **b** Plasma occludin levels in individuals with prediabetes (*n* = 20) before and 2 h after i.v. glucose infusion (paired two-tailed t test, *p* = 0.0003). Expression of **c** *Ocln* mRNA (unpaired two-tailed t test, *p* ≤ 0.0001) and **d** protein in PBS treated diabetic, db/db (*n* = 13; *n* = 5) and non-diabetic, C57BL/6j (*n* = 11; *n* = 5) mice respectively (Mann-Whitney two-tailed test, *p* = 0.0075). Expression of **e** *Ocln* mRNA (unpaired two-tailed t test, *p* = 0.0289) and **f** protein in 24 h *E. coli* infected db/db and C57BL/6j mice (*n* = 7; *n* = 6 each) respectively (Mann-Whitney two-tailed test, *p* = 0.0493). **g** Expression of *OCLN* mRNA level in TERT-NHUC uroepithelial cells cultured with glucose (normal = 6 mM: high = 30 mM) for 24 h (*n* = 13) (unpaired two-tailed t test, *p* = 0.0059). **h** Representative microscopy image of occludin after 36 h glucose treatment

(*n* = 12) (Mann-Whitney two-tailed test, *p* = 0.0034). **i** Expression of psoriasin and occludin in *S100A7* deleted (Δ*S100A7*) TERT-NHUC cells, relative densitometry of psoriasin and occludin (*n* = 8) are shown in comparison to control cells (Mann-Whitney two-tailed test, *p* = 0.0004). **j** *OCLN* mRNA after 24 h in high glucose and psoriasin (1600nM) peptide treated TERT-NHUC (*n* = 6) (unpaired two-tailed t test, *p* = 0.0011). **k** *OCLN* mRNA post 2 h TPEN treatment in TERT-NHUC, followed by a total of 24 h with high glucose treatment (*n* = 6) (unpaired two-tailed t test, *p* ≤ 0.0001). In vitro experiments were performed in duplicate or triplicate with at least 3 independent experiments and presented as mean ± SEM, statistical outliers defined by Grubb's test were excluded. For in vivo and human material analysis, median is mentioned. **p* < 0.05, ***p* < 0.01, ****p* < 0.001 and *****p* < 0.0001. Source data are provided as a source data file.

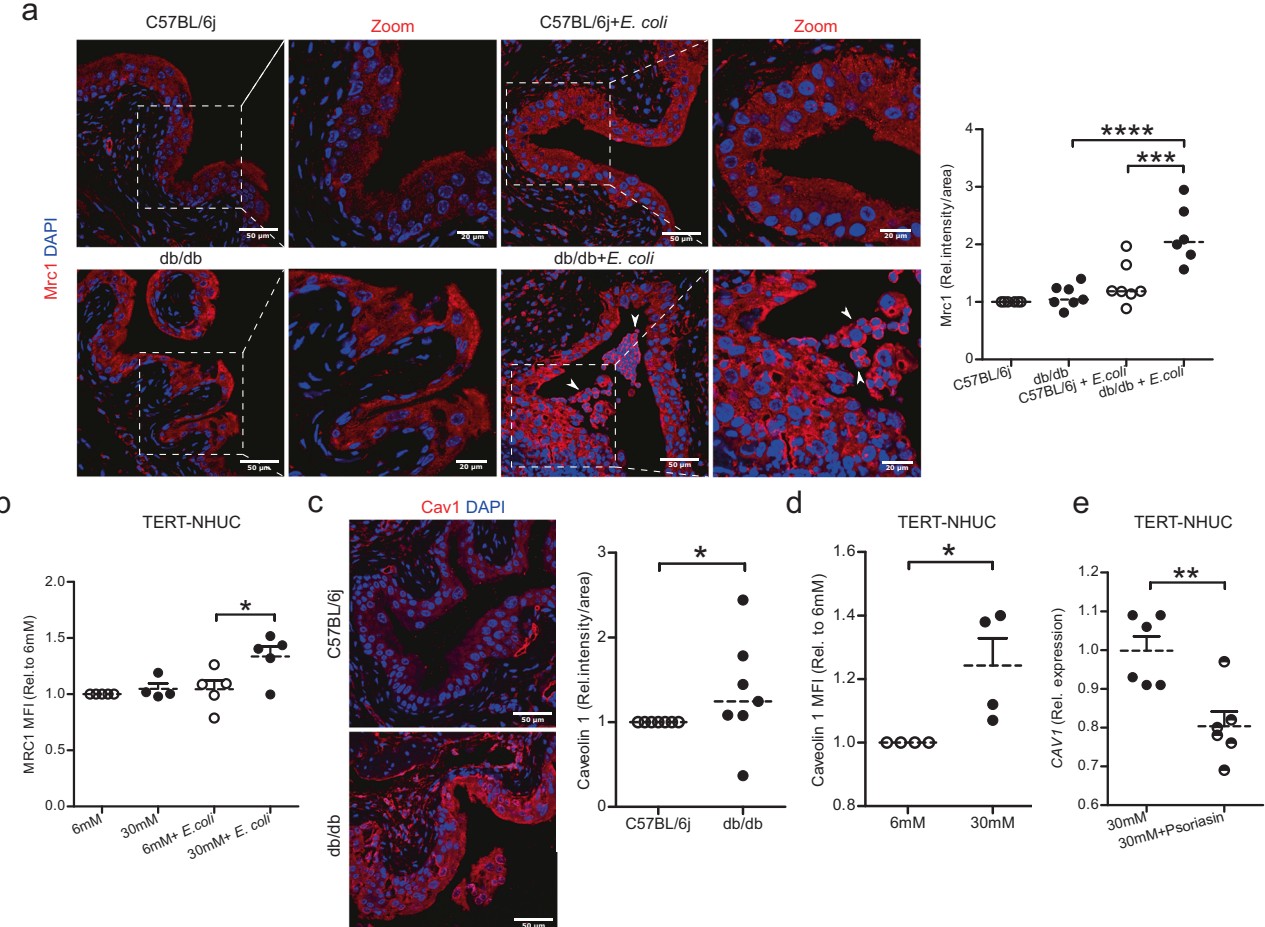

**Fig. 4 | Influence of high glucose and *E. coli* infection on MRC1 and caveolin 1.** **a** Mrc1 expression in urinary bladders from PBS treated and 24 h *E. coli* infected diabetic, db/db (*n* = 7 and 6 respectively), and non-diabetic, C57BL/6j mice (*n* = 6 and 7 respectively). Zoomed images of selected representative areas (right panel) (One-way ANOVA, multiple comparison test, $p \leq 0.001$ and $p \leq 0.0001$). **b** MRC1 levels (mean fluorescence intensity, MFI, flowcytometry) TERT-NHUC uroepithelial cells treated with glucose (normal = 6 mM: high=30mM) for 36 h, followed by 2 h infection with MOI 10, (6 mM, *n* = 5; 30 mM, *n* = 4; 6 mM + *E. coli*, *n* = 5; 30 mM + *E. coli*, *n* = 5) (One-way ANOVA, multiple comparison test, $p \leq 0.05$). **c** Urinary bladder sections of PBS treated, db/db, and C57BL/6j mice (*n* = 7 each) were stained for caveolin 1 (Mann-Whitney two-tailed test, $p = 0.0204$). **d** Analysis of caveolin 1 (MFI, flow cytometry) after 36 h of glucose treatment, (*n* = 4) (Mann-Whitney two-tailed test, $p = 0.0211$). **e** *CAV1* mRNA after 24 h in high glucose and psoriasin (1600nM) peptide treated TERT-NHUC (*n* = 6) (unpaired two-tailed t test $p = 0.0042$). In vitro experiment was performed in duplicate with at least 3 independent experiments and presented as mean ± SEM, statistical outliers defined by Grubb's test were excluded. For in vivo analysis, median is mentioned, *$p < 0.05$, **$p < 0.01$, ***$p < 0.001$ and ****$p < 0.0001$. Source data are provided as a source data file.

expression of RNase7 via the PI3K pathway[17], it is as evident from our clamp study results that insulin does not seem to exert any impact on psoriasin. We can therefore conclude that psoriasin is regulated by another pathway. This is further supported by the lack of influence by PI3K on the psoriasin expression in mammary epithelial cells, MCF-10A[35], confirming differential influence of insulin on AMPs.

In line with our clinical findings from patients with diabetes, we also detected lower psoriasin levels in the urinary bladder of diabetic mice. Previously, lower β defensin-1 was reported in diabetic rats and mice[36,37]. The importance of AMPs is evident as cathelicidin and β defensin-1 deficient mice are at greater risk for UTI[12,38]. Interestingly, *E. coli* infection of high glucose treated uroepithelial cells failed to mount an increased expression of psoriasin. Likewise, other AMPs like β defensin-3 and cathelicidin were compromised in high glucose treated keratinocytes and in macrophages, respectively[39,40].

The effect of psoriasin is particularly interesting, since it is suggested to be the most potent and abundant *E. coli*-cidal AMP[15]. RNase 7 and cathelicidin, LL-37 are 10-fold less potent than psoriasin and cathelicidin is moreover less abundant in unstimulated epithelial cells[41], suggesting them less important relative to psoriasin in

preventing *E. coli* infections. hBD-1 and mBD-1 are inactive as *E. coli* antimicrobials in their Cys-oxidized forms. Only the fully reduced, linearized form can kill *E. coli*[42]. In addition to its superior effect on *E. coli*[41], psoriasin also has a high potency against *Enterococcus* sp[41], suggesting a broader protective role against infections.

Exfoliation of infected cells is a common host driven mechanism to eliminate infection, but also allows adhered bacteria from neighboring cells to infect deeper cell layers. Increased risk of infection is often associated with loss of barrier integrity[43], allowing bacterial invasion. Tight junction proteins, distributed on epithelial cells in the urinary tract[44,45], play a key role by protecting the deeper tissue from invading pathogens.

Our findings reveal lower occludin levels in individuals with pre-diabetes after glucose infusion as well as in patients with diabetes and in diabetic mice urinary bladders. These findings were associated with lower psoriasin levels. Similarly, high glucose has been reported to reduce the expression of occludin in human retinal endothelial cells[46] and in the retina of diabetic mice[47]. We demonstrate that uroepithelial cells exposed to high glucose restored the expression of occludin after treatment with psoriasin peptide. This is also in line with psoriasin

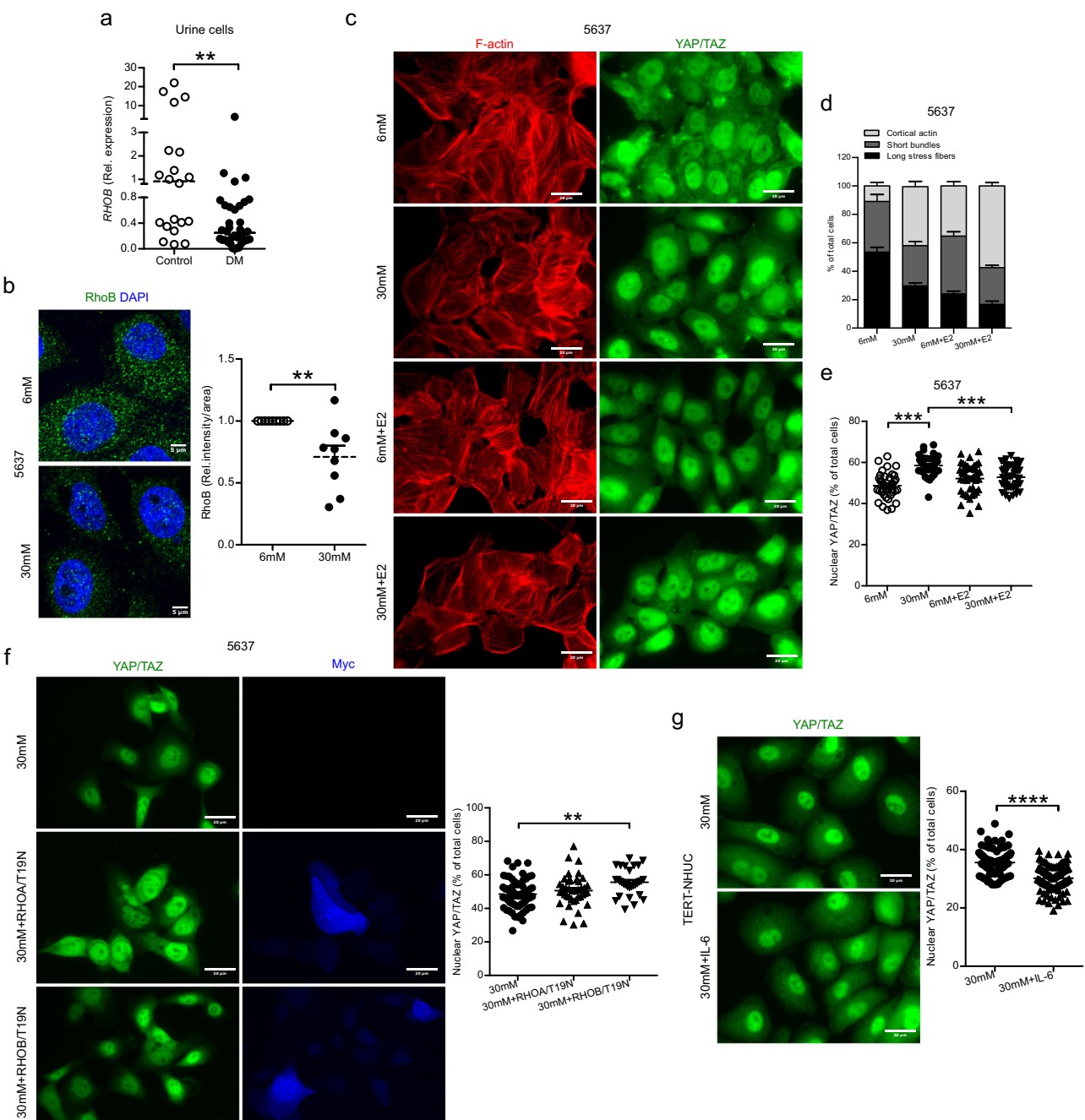

**Fig. 5 | High glucose alters the RhoB expression and translocate YAP/TAZ.**
**a** Expression of *RHOB* mRNA in urine cells from non-diabetic ($n = 20$) and diabetic patients, DM ($n = 36$) (unpaired two-tailed t test, $p = 0.0032$). **b** Microscopy analysis of RhoB was measured in 5637, uroepithelial cells after treatment with glucose (normal = 6 mM: high = 30 mM) for 36 h ($n = 9$) (Mann-Whitney two-tailed test, $p = 0.0035$). **c** 5637 treated with high glucose and estrogen (Estradiol, E2, 10 nM) for 48 h. F-actin filaments and YAP/TAZ were visualized in both normal and high glucose treated cells (6 mM, $n = 190$; 30 mM, $n = 164$; 6 mM + E2, $n = 189$; 30 mM +E2, $n = 194$), representative images are shown. **d** Quantitative analysis of cells with cortical actin, short bundles and long stress fibers is presented (6 mM, $n = 190$; 30 mM, $n = 164$; 6 mM+E2, $n = 189$; 30 mM+E2, $n = 194$) (One-way ANOVA, multiple comparison). **e** Percentage of nuclear YAP/TAZ is quantified (6 mM, $n = 40$; 30 mM, $n = 42$; 6 mM+E2, $n = 43$; 30 mM+E2, $n = 51$) (One-way ANOVA, multiple comparison, $p ≤ 0.001$). Expression of YAP/TAZ in high glucose, **f** RHOA/B T19N transfected (30 mM, $n = 61$; 30 mM+RHOA/T19N, $n = 41$; 30 mM+RHOB/T19N, $n = 30$) (One-way ANOVA, multiple comparison, $p ≤ 0.01$) and **g** 50 ng/ml of IL-6 peptide treated 5637 and TERT-NHUC cells respectively (30 mM, $n = 102$; 30 mM+IL-6, $n = 110$) (unpaired two-tailed t test, $p ≤ 0.0001$). For In vitro experiment at least 3 independent experiments were performed and presented as mean ± SEM. For in vivo and human material analysis median is shown, *$p < 0.05$, **$p < 0.01$, ***$p < 0.001$ and ****$p < 0.0001$. Source data are provided as a source data file.

mediated expression of occludin in human keratinocytes[48]. In the non-diabetic situation, other AMPs like cathelicidin and β defensin-3 were reported to trigger the expression of occludin and claudins in human keratinocytes[49,50]. Our result of restoration of *OCLN* mRNA expression in psoriasin supplemented high glucose treated uroepithelial cells

further confirmed the vital role of psoriasin in maintenance of epithelial integrity in diabetes.

In diabetes, excessive advanced glycation end products are formed and accumulate in the bladder tissues which contribute to the attachment of uropathogenic *E. coli* (UPEC) and initiation of infection[51].

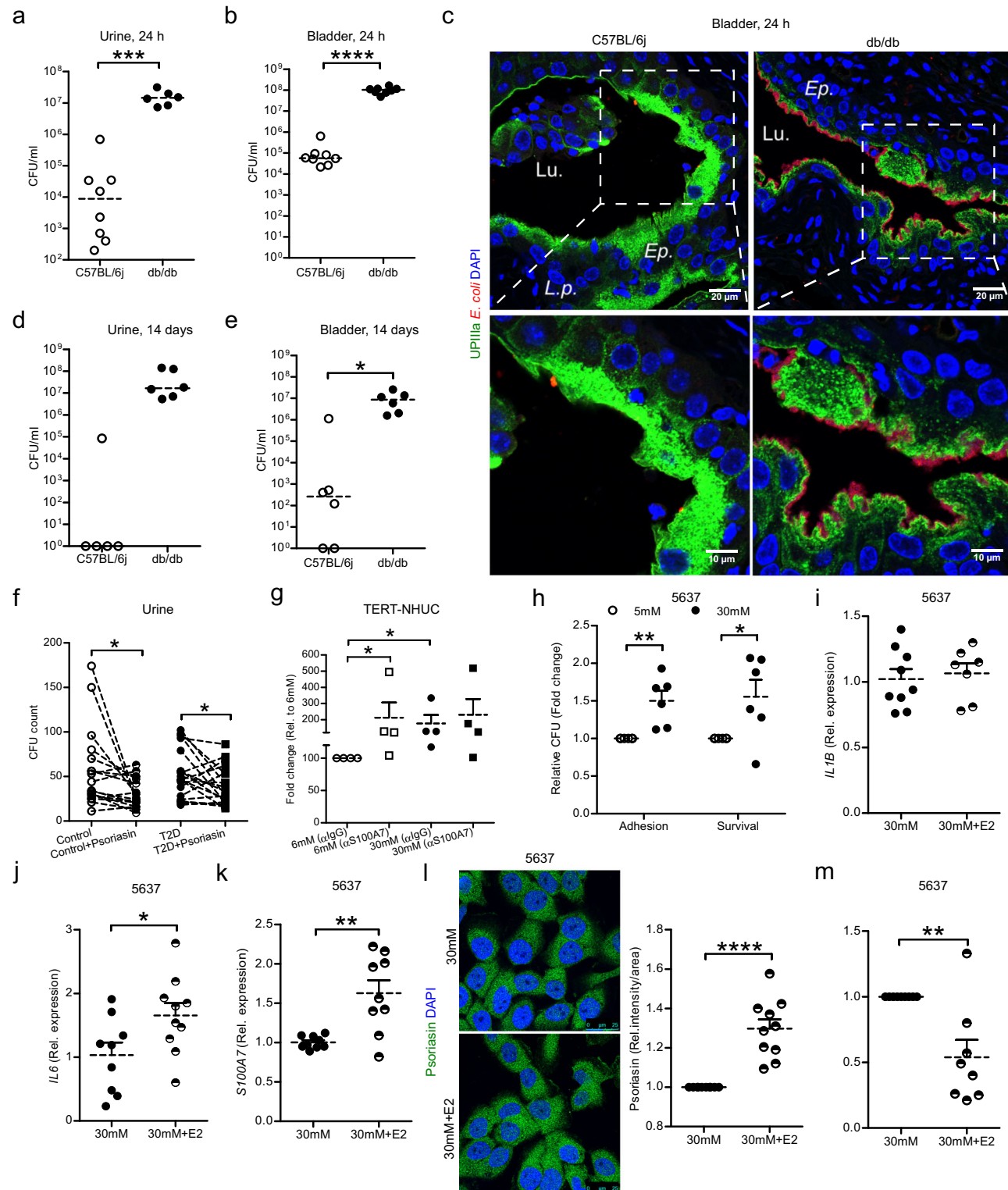

However, increased number of mannose containing binding sites for type 1 fimbriae has been demonstrated in diabetic mice urinary bladder[51]. In the current study, we demonstrate upregulation of Mrc1 in infected diabetic but not in control mice, thereby facilitating bacterial colonization and invasion, further supporting the impaired ability to control bacterial infections.

Increased expression of caveolin 1 in diabetic mice bladder creates a favorable condition for bacterial entry. Our result highlights the importance of psoriasin mediated downregulation of caveolin 1

contributing to less bacterial uptake into the cell. Rearrangement of caveolin 1 further influences the cytoskeletal backbone of the cell[52], and the organization of actin filaments in turn facilitates the ability of bacteria to multiply within the cell[53]. In addition, bacterial replication in less differentiated cell layers is inhibited by the denser actin network[54]. However, we observed that high glucose concentration triggers the formation of more cortical actin which could potentiate the development of IBCs largely in the terminally differentiated umbrella cells of diabetic mice. High glucose is known to increase the

**Fig. 6 | The impact of estradiol and high glucose on bacterial clearance.** Bacterial load in **a** urine and **b** urinary bladders of non-diabetic, C57BL/6j ($n = 8$ each) and diabetic, db/db ($n = 6$, 8 mice respectively) after 24 h *E. coli* infection (unpaired two-tailed t test, $p = 0.0003$, $p \leq 0.0001$). **c** Representative mouse bladder sections stained for UPIIIa and *E. coli* in C57BL/6j ($n = 8$) and db/db mice ($n = 7$). *L.p.*, lamina propria; *Lu.*, lumen; *Ep.*, epithelium. Bacterial load in **d** urine and **e** urinary bladders of C57BL/6j ($n = 5$ and 6) and db/db ($n = 6$ in each group) after 14 days *E. coli* infection (unpaired two-tailed t test, $p = 0.0242$). **f** Impact of psoriasin peptide (5 μM) on *E. coli* survival in urine from type 2 diabetic patients, T2D ($n = 20$) and control individuals ($n = 18$) for 30 mins (paired two-tailed t test, $p = 0.0273$, $p = 0.0442$). **g** *E. coli* survival determined in lysate of uroepithelial TERT-NHUC cells treated with glucose (normal = 6 mM: high=30 mM) for 24 h and incubated with S100A7-specific monoclonal antibodies (α-S100A7) or isotype control (α-IgG) and compared to normal glucose levels ($n = 4$) (unpaired two-tailed t test, $p = 0.0211$).

**h** *E. coli* adhesion and survival assays ($n = 6$) in 5637 after prior treatment with low and high glucose for 24 h (Mann-Whitney two-tailed test, $p = 0.0028$, $p = 0.0493$). Expression of **i** *IL1B* (6 mM, $n = 9$; 30 mM, $n = 7$), **j** *IL6* (6 mM, $n = 9$; 30 mM, $n = 10$) and **k** *S100A7* (6 mM, $n = 9$; 30 mM, $n = 9$) mRNA determined in estradiol (E2, 10 nM) and high glucose treated 5637 cells, after 48 h (unpaired two-tailed t test, $p = 0.0373$, $p = 0.0017$). **l** Expression of psoriasin protein in E2 and high glucose treated 5637 cells, after 48 h ($n = 10$) (Mann-Whitney two-tailed test, $p \leq 0.0001$). **m** Survival assay performed in 5637 cells after prior high glucose treatment with ($n = 9$) and without E2 ($n = 8$) followed by 2 h *E. coli* infection (Mann-Whitney two-tailed test, $p = 0.0058$). In vitro experiments were performed in duplicate or triplicate with at least 3 independent experiments and presented as mean ± SEM, statistical outliers defined by Grubb's test were excluded. For in vivo and human material analysis, individual values and median are shown, $*p < 0.05$ $**p < 0.01$, $***p < 0.001$ and $****p < 0.0001$. Source data are provided as a source data file.

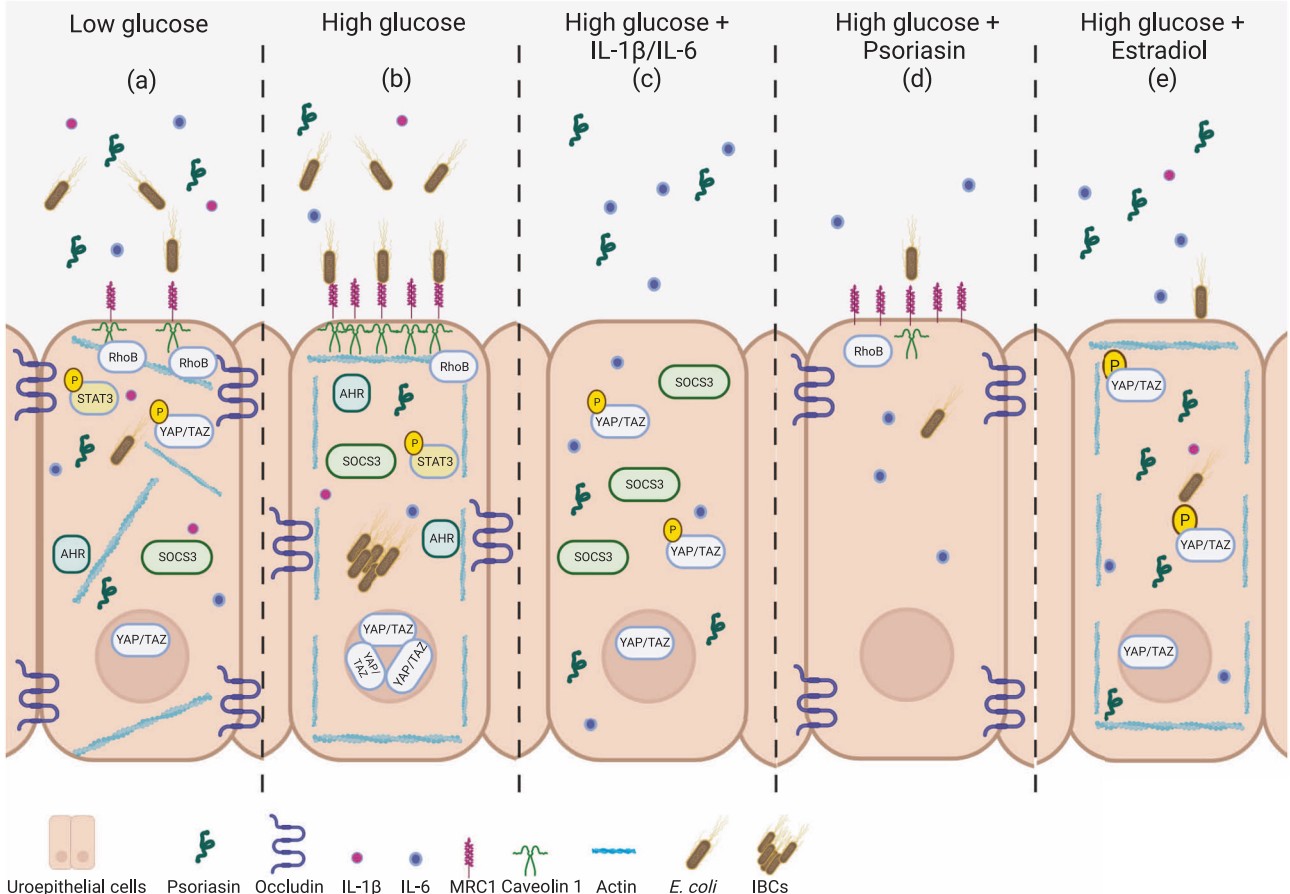

**Fig. 7 | Schematic representation of altered uroepithelial immune responses in high glucose.** The impact of high glucose on uroepithelial cells, and the effects of psoriasin, IL-1β, IL-6 and estradiol on the background of high glucose exposed uroepithelial cells is demonstrated in the current study. Immunological changes occurring due to high glucose in comparison to **a** low glucose condition. **b** High glucose significantly downregulates psoriasin, IL-1β, IL-6, occludin, SOCS3 and RhoB without altering the pSTAT-3 level, but upregulates the expression of AHR, caveolin 1 with increased nuclear YAP/TAZ and cortical actin leading to increased

bacterial load. *E. coli* infection further increases the expression of MRC1 in high glucose treated cells. **c** High glucose treated cells supplemented with IL-1β increases IL-6 and psoriasin. Supplementation of IL-6 increases psoriasin, SOCS3 and results in reduced nuclear YAP/TAZ. **d** Psoriasin peptide supplementation increases occludin and decreases caveolin 1 in high glucose treated cells. **e** Estradiol reverses the effect of high glucose and increases IL-6, psoriasin, cortical actin with reduced nuclear YAP/TAZ leading to increased intracellular bacterial killing even in high glucose treated cells.

amount of actin in pancreatic islets[55] and to cause F-actin cytoskeleton rearrangement in podocytes[56]. Intracellular filamentous actin reorganization resulted in translocation of YAP/TAZ from the cytoplasm to the nucleus[57]. This reorganization is also regulated by caveolin 1, as reported in mouse embryonic fibroblast[52]. Psoriasin expression is dependent on cell density and cell morphology[58], and nuclear YAP/TAZ significantly downregulates the expression of psoriasin in squamous cell carcinoma[32]. Therefore, our observation of glucose mediated

increased expression of caveolin 1, cortical actin formation and translocation of YAP/TAZ into nucleus emphasized a mechanistic pathway of psoriasin downregulation, confirmed in two different uroepithelial cells 5637 and TERT-NHUC. The Rho-family of small GTPases are master regulators of actin cytoskeleton rearrangements[59], and associated with enhanced formation of IBCs[60] in human bladder epithelial cells, 5637. Our results support the findings of high glucose mediated RhoB downregulation and increased intracellular bacterial

load in diabetic mice. Further, we demonstrate RhoB influencing the translocation of nuclear YAP/TAZ.

The increased bacterial load and excessive IBCs detected in diabetic mice bladders even after 14 days of *E. coli* infection indicate a lack of ability to restrict bacterial growth and distribution compared with nondiabetic controls. Further, bacteria residing in the deeper tissue can serve as reservoirs and contribute to future infections[61]. Moreover, we demonstrate the role of high glucose mediated changes in the expression of psoriasin, occludin, Mrc1 and caveolin 1 resulted in increased bacterial load. Our result therefore further adds to the understanding of increased UPEC susceptibility in prediabetic and diabetic mice model[34] and impaired UPEC clearance in diabetic mice[62].

Estradiol has recently been shown to increase glucose tolerance and insulin sensitivity in estrogen depleted ovariectomized mice[63,64]. It is also known to induce the mTOR pathway and actin polymerization through different molecular mechanisms[65]. We here demonstrate the direct effect of estradiol on *IL6* expression, a possible mechanism for an estradiol mediated psoriasin pathway without the involvement of *IL1B* in high glucose treated human uroepithelial cells. We further show that estradiol resulted in increased bacterial killing of *E. coli* infected uroepithelial cells, by translocating YAP/TAZ back to the cytoplasm and restoring the expression of psoriasin in high glucose treated uroepithelial cells without impacting the cytoskeleton. It has also been shown that estrogen receptor signaling affects psoriasin expression[66]. Therefore, we believe that estradiol has an impact on regulating the immune response. Our observation confirms the beneficial effect of estrogen also during diabetic and high glucose conditions.

We conclude that hyperglycemia compromises psoriasin through the IL-6 and YAP/TAZ pathways affecting the epithelial barrier and causing cell membrane alterations. These changes further create a favorable condition for bacterial infection (Fig. 7). Our results suggest that psoriasin in addition to other factors demonstrated by us[67] as well as others[17,34] aim to outcompete the negative consequences of high glucose. We hypothesize that these factors although partly acting independently may still be interrelated thereby strengthening the effect.

Taken together, our data suggest psoriasin as an important antimicrobial peptide in bacterial clearance of the urinary tract in diabetes and may in the future serve as a potential target for new therapeutic drugs.

## Methods
### Study participants and collection of human serum, plasma and urine exfoliated cells
The study was approved by the Regional Ethics Committee, Stockholm and performed in accordance with the Helsinki Declaration. Informed consent was obtained from all patients and volunteers participating in the study. Adult individuals with prediabetes (ethics permission 96:300), diabetes mellitus (DM) and non-diabetic controls (ethics permissions 2010/723-31/2 (amendment 2018/603-32), 2013/1618-31/3 (amendment 2014/1500-32), 2008/1804-31 (amendment 2017/477-32)) were included. Patients and volunteers with ongoing UTI, any antimicrobial treatment or estrogen supplementation, were excluded. Venous blood glucose, HbA1c and urine creatinine were analysed in Karolinska University Hospital Laboratory using standard protocol. Detailed clinical information of patients with diabetes and controls is given in supplementary Table 1. Individuals with prediabetes were not on any anti-diabetic medication and had either impaired fasting glucose (IFG), impaired glucose tolerance (IGT), or both IFG and IGT according to an oral glucose tolerance test. Both IFG and IGT confer a high risk of developing type 2 diabetes[68]. Plasma from patients studied during euglycemic hyperinsulinemic clamps (ethics permission 2009/623-32) and prediabetes individuals during hyperglycemic clamps were included and have been reported earlier[18,19].

### Bacterial strain
Uropathogenic *E. coli* strain CFT073 was used for in vitro and in vivo experiments. This strain was isolated from a patient with acute pyelonephritis and expresses type 1, P and S fimbriae along with α-hemolysin. Bacteria were grown over night on blood agar plates at 37° C followed by 4 h in Luria-Bertani broth to reach the logarithmic phase[69]. Bacteria were washed twice with phosphate-buffered saline (PBS) and bacterial concentration was measured spectrophotometrically and confirmed by viable count.

### Mouse model of UTI
Mice experiments were approved by the Northern Stockholm Animal Ethics Committee (animal ethics permission N-177/14 (amendment 10370-2018)), and experiments were carried out according to the guidelines of the Federation of Laboratory Animal Science Association and in compliance with the Committee's requirements. Eight-week-old female db/db (BKS (D)-*Lepr*db/JOrlRj) with type 2 diabetes (median blood glucose: 17.1 mmol/l) and wildtype C57BL/6j mice (median blood glucose 7.1 mmol/l) were obtained from Janvier Laboratories. All mice were kept in a specific pathogen free facility in individual ventilated cages with aspen bedding housing 4 mice per cage. A 12 h light, 12 h dark cycle in ambient room temperature and humidity was maintained, with food and water ad libitum. At week 10, infection was performed. Water was withdrawn 4 h prior to bacterial inoculation, mice were anaesthetized using isoflurane and transurethrally infected with $0.5 \times 10^8$ colony-forming units (CFU) of *E. coli* CFT073 in 50 μl of PBS, after which water was returned[70]. Blood glucose levels for db/db and C57BL/6j were measured before infection, during the course of infection and at sacrifice after 24 h, 7 or 14 days of infection. Urine was collected and respective urinary bladders were aseptically removed, cut open and washed with PBS to remove urine and non-adherent bacteria. To determine the total bacterial load, adhered and intracellular bacteria, bladders were homogenized in 1 ml of PBS, serially diluted and both bladders and urine were plated on blood agar plates.

### Cell lines and culture conditions
Telomerase-immortalized human uroepithelial cells, TERT-NHUC (kindly provided by M. A. Knowles, Leeds, UK) and 5637 (HTB-9, American Type Culture Collection) were cultured as previously described[70]. TERT-NHUC cells were grown in EpiLife medium (6 mM of glucose, Life Technologies) supplemented with 1% of human keratinocytes growth supplement (HKGS, Life technologies) and 5637 cells in RPMI 1640 medium with no glucose (Life Technologies) but supplemented with 5 mM glucose (Sigma) and 10% fetal bovine serum (Life Technologies) and cultured at 37°C and 5% $CO_2$. To mimic hyperglycemia, cells were exposed to 11 and 30 mM of glucose for 24 to 72 h as appropriate, while normoglycemia, 6 mM, was obtained with culture media for TERT-NHUC while 5637 cells were supplemented without cytotoxicity or compromising proliferations (Suppl. Fig. 1h, i respectively). For experiments with estradiol, phenol red–free medium and 5% charcoal–treated fetal bovine serum were used. 17β-estradiol (Sigma) in absolute ethanol was used at a final concentration of 10 nM. Cells were treated for 48 h, with medium exchanged after 24 h. For experiments with Latrunculin B (Lat B, Sigma), cells were treated with 1 μM of Lat B 1 h prior to adding glucose and estradiol. Infection was initiated after 24 h. During intracellular zinc chelation experiment, 20 μM of N,N,N′,N′-tetrakis (2-pyridylmethyl) ethylenediamine, (TPEN; Sigma) was added to human uroepithelial cells, TERT-NHUC for 2 h in minimal essential medium (MEM; Life Technologies), followed by treatment with high glucose for a total 24 h in Epilife medium.

### Cell infection assays
Cell experiments were carried out in 24 well cell culture plates, human uroepithelial cells TERT-NHUC in Primaria (BD) and 5637 cells in Costar plates. Cells were grown in the presence of 5 or 6, 11 and 30 mM

glucose, respectively. All media were antibiotic and serum free. A total $10^6$ CFU/ml (MOI 5) or $2 \times 10^6$ CFU/ml (MOI 10) of *E. coli* CFT073 were added to nearly confluent pre-treated cells and incubated in 37 °C at 5% $CO_2$ and 80% humidity. At time points 5, 15, 30, 60 and 120 mins, cells were washed once with PBS and harvested for further analysis.

## Adhesion and survival assay

TERT-NHUC and 5637 cells were infected with $10^6$ *E. coli* in 100 μl of PBS per well, centrifuged for 1 min at 350 g. For assessment of bacterial adhesion, cells were infected for 30 mins only and washed with PBS. In survival assays, cells were washed with PBS after 2 h infection to remove non-adherent bacteria and supplemented with fresh medium for another 2 h. In hyperglycemic cells, medium was supplemented with 30 mM glucose throughout the entire experiment. At indicated time points, cells were lysed with 0.1% Triton-X-100 in PBS (pH-7.4) and serially diluted and plated on blood agar plates. Rate of adhesion and survival were calculated by number of adhered or intracellular bacteria in relation to the total number from the same experiment.

## Antimicrobial activity, effect of psoriasin

Human uroepithelial cells, TERT-NHUC were treated with 6 or 30 mM glucose. After 24 h of treatment, medium was removed, and cells were lysed in 1% Triton X-100 in PBS. Cell free supernatant was obtained by centrifugation at 8000 g for 5 mins and then incubated for 30 mins at 37 °C with 1 μg/ml of monoclonal mouse anti-psoriasin antibody (Santacruz Biotechnology) or the same concentration of an isotype control antibody (mouse $(Ig)G_1,\kappa$; BD Biosciences). Bacteria were prepared as above and 50 μl from $10^4$ CFU/ml bacterial suspension was added to 150 μl of pretreated cell lysate. After incubation for 30 min at 37 °C, 100 μl aliquots were plated and bacterial survival was determined by viable count. Results were expressed in relation to control cell lysates pretreated with control antibodies.

Urine samples, with no bacterial growth from nondiabetic controls and T2D patients were collected, and *E. coli* CFT073 was prepared as above and 50 μl from $10^4$ CFU/ml bacterial suspension was added to 150 μl of urine with 5 μM of psoriasin peptide. After incubation for 30 min at 37 °C, 100 μl aliquots were plated and bacterial survival was determined by viable count.

## Total RNA isolation and real-time PCR

After the required incubation, human uroepithelial cells were directly processed for RNA isolation, whereas bladder tissues were homogenized manually with Dounce's homogenizer. Total RNA was extracted using the RNeasy Mini kit (Qiagen) according to the manufacturer's protocol. The concentration and purity of RNA was determined with nanodrop, and up to 0.5 μg of RNA was reverse transcribed using random primers for 10 min at 25 °C, 120 min at 37 °C, and inactivation at 85 °C for 5 min and was transcribed to cDNA using the High-Capacity cDNA Reverse Transcription Kit (Applied Biosystems) in thermal cycler, MJ research, PTC-200. Real-time PCR was performed after initial denaturation at 95 °C for 10 min, each cycle consisted of 15 s at 95 °C, 60 s at 60 °C (touchdown of 1 °C per cycle from 66° to 60 °C), and 30 s at 72 °C using standard SYBR green (Applied Biosystems) or using probes in TaqMan gene expression assays (Applied Biosystems) for both human and mouse specific genes in a Rotor-Gene PCR cycler (Corbett Life Science, RG 3000, rotor gene version 6.1). All the primers and probes used in this study are mentioned in supplementary table 2. Relative expressions of target genes were presented as $2^{-\Delta CT}$ and fold change as $2^{-\Delta\Delta CT}$ compared to uninfected or non-treated control.

## Immunofluorescence of bladder sections and cells

Mice experiments were performed as described earlier. At the required time point, bladder tissue was fixed in 1 ml of 4% PFA for at least 36 h, and then transferred to 1 ml of absolute ethanol. Paraffin blocks were prepared and 4 μm sections were cut using microtome. Sections of paraffin-embedded mouse bladder tissue were deparaffinized and rehydrated, pretreated with 0.3% Triton X-100/PBS at room temperature, or boiled in citrate buffer, 1 mM EDTA, 10 mM tris, 0.05% Tween 20 (pH 9); for psoriasin (S100a7a), IL-1β, IL-6, mannose receptor c-type 1 (Mrc1; CD206), caveolin 1 (Cav1), occludin (Ocln) and Rhob staining. TERT-NHUC, human uroepithelial cells were fixed in 4% PFA for 15 mins at room temperature and permeabilized with 0.3% Triton X-100 in PBS. Thereafter, sections were blocked for 30 mins with FX Signal Enhancer (Invitrogen), and both cells and sections were blocked for an additional 60 mins with the sera from the species in which the secondary antibodies were raised. Incubation with primary antibodies was carried out overnight at 4 °C followed by secondary Alexa Fluor–conjugated antibodies for 1 h at room temperature. Antibodies used were goat anti *E. coli* (1:200, AbD Serotec; chicken anti goat Alexa Fluor 594 or 488 (1:600), Invitrogen), rabbit anti uroplakin IIIa (UPIIIa, 1:200, Santacruz; donkey anti rabbit Alexa Fluor 488 (1:600), Invitrogen), mouse anti psoriasin (1:200, Santacruz; rabbit anti mouse Alexa Fluor 488 (1:600), Invitrogen), mouse anti occludin (tissue: 1:200, Santacruz; rabbit anti mouse Alexa Fluor 594 (1:600), Invitrogen) or rabbit anti occludin (cells: 1:200, Invitrogen; donkey anti rabbit Alexa Fluor 488 (1:600), Invitrogen), mouse anti caveolin 1 (1:200, Santacruz; rabbit anti mouse Alexa Fluor 594 (1:600), Invitrogen), mouse anti mannose receptor c-type-1 (CD206, 1:100 Abcam; rabbit anti mouse Alexa Fluor 594 (1:600), Invitrogen), rabbit anti IL-1β (1:200, Invitrogen; donkey anti rabbit Alexa Fluor 488 (1:600), Invitrogen), rabbit anti IL-6 (1:100, Invitrogen; donkey anti rabbit Alexa Fluor 594 (1:600), Invitrogen), TRITC-labeled phalloidin (1:350; Sigma), Phalloidin Alexa Fluor 488 (1:1000; Invitrogen), rabbit anti RhoB (1:200, Invitrogen; donkey anti rabbit Alexa Fluor 488 or 594 (1:600), Invitrogen) and YAP/TAZ (1:100, Santacruz; rabbit anti mouse Alexa Fluor 488 (1:500), Invitrogen). Sections and cells were then mounted with ProLong Gold Antifade mounting medium including DAPI (Invitrogen). For nuclear YAP/TAZ estimation, the integrated density module in ImageJ was used to measure the total fluorescence intensity of each cell, followed by the fluorescence intensity in the nucleus of each cell. The proportion of nuclear over total cellular YAP/TAZ was calculated. Analysis of the type of actin filament organization was done by analyzing images acquired by immunofluorescence microscopy for the presence of short actin bundles, stress fibers or cortical actin as the dominant form of actin organization. Quantification was performed from three independent experiments per experimental condition. Imaging was performed with Leica SP5, Zeiss LSM 700 confocal microscopes and Zeiss AxioVert 40 CFL epifluorescence microscope, fluorescence intensity per unit area were analyzed in image J Fiji 1.53b software.

## Psoriasin, occludin and IL-6 ELISA

Serum and plasma from patients with diabetes, along with hyperglycemic clamped pre-diabetic and hyperinsulinemia clamped T1D and non-diabetic control samples were collected before and after 2 h of glucose or insulin treatment respectively. Supernatants from 24 h glucose treated TERT-NHUC cells were collected and centrifuged at 350 g for 10 min and stored at −80 °C until assayed. ELISA were analyzed using the CircuLex S100A7/Psoriasin ELISA Kit (MBL International) and IL-6 (R&D Biosystems) according to the manufacturer's recommendations in EZ400 microplate reader (Biochrom) using ADAP software version 2.0. Uninfected cells and serum and plasma obtained from nondiabetic individuals and before glucose and insulin infusion served as controls.

## Flow cytometry

To investigate the protein expression of psoriasin, occludin, and caveolin 1, TERT-NHUC cells were harvested after 36 h glucose treatment. For MRC1 (CD206), cells were further infected for 2 h, centrifuged at 350 g for 3 min at room temperature (RT). 1 ml of 4% PFA in PBS (Fisher Scientific) was added to the cell pellet. Cells were

incubated at RT for 15 mins, centrifuged and 1 ml of 0.1% Triton-X-100 in PBS (PBST) was added and incubated in RT for 10 min. Thereafter cells were blocked with 5% BSA for 30 min, stained with primary antibody in 1:1 ratio of 200 μl of 1× PBST and 5% BSA for 30 mins at RT. After primary antibody staining, cells were washed with 1× PBS with 1% BSA and further incubated with respective secondary Alexa fluor 488 (1:600, Invitrogen) or Alexa fluor 647 (1:400, Invitrogen) antibodies in 1:1 ratio of 200 μl of 1× PBST and 5% BSA for 25 min in dark at RT. Antibodies used are rabbit anti pSTAT-3 (1:100, Cell Signaling Technology; donkey anti rabbit Alexa Fluor 488 (1:400, Invitrogen), mouse anti STAT-3 (1:100, Cell Signaling Technology; goat anti mouse Alexa Fluor 647 (1:400, Invitrogen), mouse anti psoriasin (1:200, Santacruz; rabbit anti mouse Alexa Fluor 488 (1:400), Invitrogen), mouse anti caveolin 1 (1:200, Santacruz; rabbit anti mouse Alexa Fluor 488 (1:400), Invitrogen), and rabbit anti mannose receptor c-type 1 (1:200, Invitrogen; donkey anti rabbit Alexa Fluor 488 (1:400), Invitrogen). Finally, cells were dissolved in PBS and data acquired on a BD LSRFortessa™ and analyzed in FlowJo software version 10.8.1. The gating strategy used standard FSC and SSC, indicating boundaries between positive and negative cell populations. Representative contour plot for each protein of interest are shown in Supplementary Fig. 5.

## S100A7 deletion using crispr/cas9 system

Human uroepithelial cells TERT-NHUC were cultured in 24-well plates to 50–70% confluency and transfected with non-targeting SpCas9/gRNA (Synthego) or a pool of *S100A7* specific SpCas9/gRNA (Synthego) using Lipofectamine™ CRISPRMAX™ Cas9 Transfection Reagent (Thermo-Scientific) according to the instructions of the manufacturer. After transfection the TERT-NHUC cells were cultured for 72 h and then fixed and processed for microscopy analysis of mouse anti psoriasin (1:200, Santacruz; rabbit anti mouse Alexa Fluor 488 (1:600), Invitrogen) and rabbit anti occludin (1:200, Invitrogen; goat anti rabbit Alexa Fluor 647 (1:600), Invitrogen). Pooled *S100A7* gRNA: #1 A*C*A*CAGGCACUAAGG AAGUU, #2 A*C*A*CAGGCACUAAGGAAGUU, #3 A*U*U*UUUUAAUCAG AGGGUGA. Non-targeting gRNA: G*C*A*CUACCAGAGCUAACUCA. The gRNA had chemically modified scaffolds (Synthego).

## RHOA/B transfection

Human uroepithelial cells 5637 were cultured in 24 well plate. When 70% confluency was reached, cells were pretreated with high glucose for at least 6 h prior to the transfection. 1 μg of Myc-tagged RHOA/T19N and RHOB/T19N were separately prepared in 150 mM NaCl to a final volume of 50 μL. 2 μL of jetPEI® reagent in 150 mM NaCl to a final volume of 50 μL. 50 μL jetPEI® solution was added all at once into the 50 μL DNA solution and incubated for 20 min at room temperature. 100 μL jetPEI®/DNA mix was added drop wise to the cells in 1 ml of serum-containing medium and homogenized by gently swirling the plate. Cells were incubated for 24 h followed by fixation with 1 ml of 4% PFA and were processed for microscopy analysis of mouse anti YAP/TAZ (1:100, Santa Cruz) detected by rabbit anti mouse Alexa Fluor 488 (1:500, Invitrogen) conjugated antibody and Myc-tagged RHOA/T19N and RHOB/T19N detected with rabbit anti Myc antibody (1:200, Sigma Aldrich) followed by donkey anti rabbit Alexa Fluor 350 (1:500, Invitrogen) conjugated antibody using standard imaging protocol.

## Peptides

Purified $Zn^{2+}$ free, natural skin derived, 11,366-Da psoriasin peptide[15] and recombinant human IL-1β (Miltenyi Biotec) and IL-6 (Invitrogen) were used to stimulate TERT-NHUC for 24 h. The concentrations used were 1600 nM or 5 μM, 20 ng/ml and 50 ng/ml respectively. IL-1β blocking was achieved by addition of IL-1β specific inhibitor, diacerein (Sigma, 50 μM) for at least 4 h prior to the IL-1β peptide treatment followed by overnight treatment in high glucose treated TERT-NHUC.

## Statistical analysis

All statistical tests were performed in Graph pad Prism version 5. No samples were excluded from human and animal studies. For in vitro analysis using human uroepithelial cells, statistical outliers defined by Grubb's test were excluded. Data were obtained from Students unpaired t-test, non-parametric test using Mann Whitney U test, paired Students t-test and non-parametric one-way ANOVA, with Bonferroni's or Dunnett's multiple comparison tests as appropriate. Differences with p values below 0.05 were considered statistically significant.

## Reporting summary

Further information on research design is available in the Nature Research Reporting Summary linked to this article.

## Data availability

Our data do not mandate deposition in a public repository. All raw files and other relevant information are stored in the Karolinska Institutet's Electronic Lab Notebook. As the Karolinska Institutet's Electronic Lab Notebook is not a public repository, information may be provided from the corresponding author upon reasonable request. Source data are provided with this paper as a Source Data file.

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

## Acknowledgements

We thank all the volunteers for their participation in the study. Staff at Karolinska University Hospital, Danderyd Hospital, South Hospital and Capio Health Care Center are acknowledged for help collecting samples. We thank John Kerr White for drawing the graphical summary Fig. 7

and the human objects in Fig. 1c, d. These figures were made using a paid subscription with BioRender.

## Author contributions

S.M. and A.B. conceived and designed the experiments; S.M., W.K., A.S., P.A. performed research; A.B.j., J.T., K.B., T.N., C.-G.Ö., H.B. contributed human samples; A.E., J.S., C.-G.Ö., H.B., A.B. provided reagents/ new analytic tools; S.M., W.K., A.S., C.-G.Ö., P.A., H.B. and A.B. analyzed data; S.M. and A.B. wrote the paper; A.B. supervised the work. All authors read and approved the manuscript.

## Funding

This research was funded by the Stiftelsen Olle Engkvist Byggmästare, Region Stockholm (ALF project) and Swedish Neurological Association (A.B.). Karolinska Institutet's Research Foundation (S.M., W.K., and A.B.), W.K. was supported by grants from Siriraj Hospital Mahidol University, Thailand. H.B. is supported by grants from the Swedish Society for Medical Research, the Swedish Cancer Foundation, the Swedish Medical Association, Region Stockholm (clinical research appointment), Hudfonden, Clas Groschinsky, Åke Wiberg, Magnus Bergvall and Karolinska Institutet foundations. Open access funding provided by Karolinska Institutet.

## Competing interests

The authors declare no competing interests.
