## [Peer Review File · Nature Communications]

Diabetes downregulates the antimicrobial peptide psoriasin and increases E. coli burden in the urinary bladderReviewers' Comments:

Reviewer #1:

Remarks to the Author:

The report by Mohanty et al describes the impact of diabetic hyperglycemia on the innate immune defenses of the urinary tract. It is well known that diabetics are prone to serious urinary tract complications. Recent studies have supported the hypothesis that UTIs occurs due to a failure of AMP expression, rather than as a consequence of nutrient-rich urine that supports microbial growth. This report focuses on psoriasin. The basic message of the report is that high blood glucose suppresses expression of psoriasin. This is demonstrated elegantly using glucose clamped subjects, prediabetics, controls, and db/db mice. The authors show that, in turn, reduced psoriasin leads to increased bacterial growth in urine, and reduced bacterial clearance in vivo. The authors further show that high concentrations of glucose stimulate the translocation of YAP to the nucleus, a transcription factor involved in repression of psoriasin. In addition, elevated glucose concentrations induce the mannose receptor, which facilitates microbial adherence and represses the expression of occludin, which facilitates subepithelial microbial invasion. In addition elevated glucose inhibited expression of IL1b and IL 6, both involved in expression of psoriasin. Curiously, estradiol stimulates psoriasin expression in the presence of high glucose concentration.

The manuscript is well written and the data clearly presented. I found it useful that the authors included several responses to elevated glucose that predispose to bacterial infection, in addition to effects on psoriasin. Most important, from my perspective, is that adds to our understanding of the impact of high urinary concentrations of glucose on the risk of urinary tract infection. I have no significant criticisms.

Reviewer #2:

Remarks to the Author:

Nature Communications NCOMMS-21-38134

The present study describes the influence of hyperglycemia on psoriasin expression in the urinary bladder and whether it affects anti-infective response to Escherichia coli.

The authors analyze both patient's samples and mouse models to demonstrate that hyperglycemia decreases psoriasin expression and compromised the response to E. coli.

The study shows:

-In hyperglycemic patients and mice, psoriasin level is reduced in urine, serum, and bladder and this reduction was dependent on glucose and not insulin.

-Similar conclusions were obtained in vitro with uroepithelial cell line.

-High glucose decreases occluding expression in mouse bladder and in uroepithelial cell line and this decrease is rescued by addition of psoriasin.

-High glucose increases MRC1 and caveolin 1 expression after E. coli infection in mice and uroepithelial cells favoring bacterial attachment.

-In uroepithelial cell line, high glucose decreases expression of RHOB a factor regulating actin assembly and increases nuclear YAP a factor downregulating psoriasin expression. Estradiol reverses this phenotype.

-High glucose decreased IL-6 expression via blocking IL-1b expression, explaining the decreased expression of psoriasin known to be induced by IL-6.

-Diabetic mice showed impaired clearance of bacterial infection.

-Uroepithelial cell line infected with E. coli showed increased bacterial load after culture with high glucose and this phenotype is rescued by estradiol that increased psoriasin.

The present study may provide a mechanism explaining the observation that hyperglycemic patients have higher risk of recurrent bacterial infection. While interesting the study remains very puzzling and it is not obvious whether every data provided are connected to each other. Also, while one strength of

the study is the use of both patient's samples and mouse model, the authors did not take advantage to the mouse model to clearly demonstrate the mechanism linking glucose to psoriasin and bacterial infection. Finally, the study is focused only on psoriasin and the redundancy with the other antimicrobial peptides (cathelicidin, mBD1) of which the expression is also decreased by glucose, is not considered.

Specific points:

-Figure 1: T1D and T2D patients should not be mixed since many other parameters additionally to hyperglycemia may influence psoriasin level in T1D patients. In E-G, both groups (B6, db/db mice, uninfected, infected) should be shown together on the same graph.

What is the significance of lower level of psoriasin observed in hyperglycemic condition in uninfected context? What is the level of expression of other AMPs (cathelicidin, mBD1)?

-Figure 2: What is the mechanism of induction of occludin by psoriasin? Decreased in occludin expression is directly mediated by glucose and/or via decreased of psoriasin?

-Figure 3: The observation on MRC1 and caveolin-1 expression is interesting, however what is the link with psoriasin? What is the link between RhoB and YAP localization? The effect of estradiol is interesting however what is its mechanism of action?

-Figure S4: why these data that provide some mechanisms are in supplementary? What would be the source of IL-1b in vivo? The uroepithelial cells? In vitro, does high glucose decreases IL-1b expression by uroepithelial cells and consequently IL-6 and psoriasin?

-Figure 4: these in vivo data should be presented earlier in the manuscript. However additional experiments should be performed: treatment of infected diabetic mice with psoriasin and estradiol. In vitro, whether psoriasin is more effective than other AMPs produced by uroepithelial cells to kill E. coli.

A graphical abstract should be very useful to summarize and link the different data described thorough the study.

Reviewer #3:

Remarks to the Author:

This manuscript by Mohanty et al, investigates how diabetes alters innate immune defenses within the context of E. coli cystitis, a common infection in those with diabetes. The authors specifically focus on the antimicrobial peptide psoriasin (S100A7). They use 3 different models of study – samples from patients with or without diabetes (including those treated with hyperglycemic or euglycemic clamps), a mouse model of type 2 diabetes, and 2 uroepithelial cell lines. They found that high glucose concentrations induced lower psoriasin levels and impaired epithelial barrier through altered cytoskeletal function. These changes resulted in increased bacterial burdens including increased amounts of intracellular bacterial communities within bladder epithelial cells. Interestingly, exogenous replacement of psoriasin rescued some of these cytoskeletal abnormalities, and estradiol treatment led to enhanced levels of psoriasin. These findings will be of interest to those interested in host-pathogen interactions in the context of metabolic disease, which is becoming a very common risk for infections. This paper is strong in several respects including its use of patient samples derived during the hyper/euglycemic clamps and use of multiple models to verify their findings. The methodology is appropriate to the studies conducted and the results and interpretation are clearly presented. This Reviewer's enthusiasm for this otherwise strong manuscript is slightly lessened by a story that in places reads more as an unrelated list of immunologic abnormalities related to DM rather than a clear mechanistic change that drives these abnormalities. For instance, the connection between Mrc1, RhoB, caveolin-1 in relation to psoriasin is not made clear. Are the authors suggesting a series of cellular changes that culminate in psoriasin expression or a convergence of several different immunologic

abnormalities that contribute to the phenotypes of infection? Are the functions of YAP and IL-6 inter-related or just alternative ways to regulate psoriasin expression? The estradiol findings are intriguing but it is unclear why this was tested, and it is not clear if estradiol provides broad protective effects via changes in glucose tolerance or triggers a specific pathway through psoriasin. The manuscript also felt like it ended abruptly without further investigating the in vivo effect of estradiol (or IL-6) on diabetic mice challenged with UTI.

Major Critiques

1. How was the percent of cells with each type of actin filaments and quantification of nuclear vs cellular YAP determined? This does not appear to be described in the methods.
2. The findings in figure 3 could use additional text to bridge connections between the cellular changes that authors identified and the consequences for the host-pathogen interaction they are examining. The authors seem to be suggesting that the upregulation of Mrc1 and caveolin would increase adhesion and uptake by e.coli (potentially resulting in the IBCs seen in figure 4). What is the role of RhoB downregulation? Is this speculated to enhance IBC formation? Are the findings of RhoB downregulation connected to the actin filaments and YAP nuclear translocation? In the discussion the authors speculate that high glucose concentrations triggers formation of more cortical actin which could potentiate the development of IBCs, but in figure 3G, estradiol did not improve the amount of cortical actin, but still resulted in less CFU in Figure 4K. How are the findings between relative CFU and actin structure related in this context?
3. Consider better summarizing if the authors think their findings represent a series of cellular changes that culminate in psoriasin expression or a convergence of several different immunologic abnormalities that contribute to the phenotypes of infection. A summary figure may be helpful in this regard.

Minor critiques

1. Data regarding Mrc1 is actually present in Figure S2c and S2d (not S3c/d) as listed in the text (lines 160 and 163).
2. Figure 3B needs to be larger, even on 150% magnification, the labels are barely readable. If needed, I would recommend removing the histogram and just showing the mean values as depicted in the insert.
3. Figure 3F - the merged images of actin (red), dapi (blue), and YAP (green) IF staining are presented. The green staining (nuclear vs cellular) is difficult to discern due to the dapi staining, consider adding separate color channel images to supplementary figure 3.
4. In Figure S2, for the db/db sample, there is no visible red staining, though the quantification suggest the occluding staining should be similar to that seen in Figure 2D or F? Does this image include the red channel?

Reviewer #4:

Remarks to the Author:

Summary and General Comments: This paper by Mohanty et al concerns the effects of glucose on the anti-microbial peptide psoriasin and E. coli burden in the urinary bladder. They find that high glucose and not insulin reduces the psoriasin levels and impairs epithelial barrier function with altered cell membrane proteins and cytoskeletal organization. This was associated with increased bacterial burden reflected by increased intracellular bacterial communities. They show the high glucose impairs the release of proinflammatory cytokine IL-6 and IL1B which they postulate affects the psoriasin levels. Estrogen treatment restored the effects of hyperglycemia. They conclude that the high glucose seen in diabetes underlies compromised innate immunity in the bladder.

This is an interesting study that is well focused on innate immune mechanisms in the bladder. The authors suggest that the findings may explain the relationships between high glucose and impaired immunity that leads to urinary tract infections. There are a few specific questions about the data presentation, described below, but overall it is well presented. The conclusions that are drawn are

supported by the presented data. A major limitation of the work is the absence of a clear mechanism whereby glucose affects the innate mediators. They suggest that IL-6 may be an important intermediate but the way in which IL-6 does this (through STAT3??) and how glucose affects IL-6 levels are not studied.

The significance of the findings and relevance to disease is not clear. Based on their proposition, there should be a relationship between glucose control (measured by HbA1c levels) and psoriasin or S100A7 levels in the urine and urine cells as shown in Figures 1a, b. It would seem that this question should at least be addressed since the investigators have the data (Suppl Table 1). In addition, they perform a euglycemic hyperinsulinemic clamp in patients with T1D and do not find a difference in the psoriasin levels and claim that insulin does not affect the psoriasin levels. However, the studies could have more precisely addressed this question by a euglycemic or hyperglycemic clamp with the same insulin delivery in patients with T1D who do not make their own insulin. Likewise, clamp studies could be done in rodents to answer the question about the relationship between insulin and psoriasin. More challenging to understand is the timing of the changes in the psoriasin levels and changes in glucose. They describe changes that occur within minutes. It leaves the reader to wonder if the changes they describe (Figure 1i) are of significance. Finally, the restoration of the effects of hyperglycemia with estrogen also raise question about the clinical significance. Women with diabetes are more than twice as likely at non-diabetic women to have a urinary tract infection and these infections are more common in women than men.

Specific points:

1. A common statistical approach that is used is to look at fold change with an experimental condition compared to a baseline. Presenting the data in this way (eg. Fig 2i, Fig S5e) does not allow us to see the variance in the data at the baseline. Furthermore, some values increase and some decrease and the relationship between the baseline value and the change cannot be appreciated in the way this data are presented. It appears that what they actually did is a 1 sample t-test comparing the fold change to 1. The statistical approach should be clarified.
2. Fig1B: How did body weight affect the psoriasin concentrations? What is the appropriate control group?
3. Fig 1C – the changes in the plasma are modest despite the statistical significance.
4. In Fig S2, db/db mice are compared to B6 mice (which are all assigned 1). However, these are two different strains so a paired test here is not appropriate.
5. Throughout the paper relative mRNA levels are presented. However, in the methods, RT-PCR is described (line 456). How was mRNA measured? If it was RT-PCR, what was the reference cDNA and what data are presented.
6. Fig S1 – psoriasin is not different in DM vs non-diabetic controls??? Where is this measured?
7. In Fig S5e: We don't know the baseline levels (at 6mM). If fold levels are to be compared, why are the 6 mM data presented or if they are presented we should see the actual values.

Responses to Reviewers.

Reviewer #1 (Remarks to the Author):

The report by Mohanty *et al* describes the impact of diabetic hyperglycemia on the innate immune defenses of the urinary tract. It is well known that diabetics are prone to serious urinary tract complications. Recent studies have supported the hypothesis that UTIs occurs due to a failure of AMP expression, rather than as a consequence of nutrient-rich urine that supports microbial growth. This report focuses on psoriasin. The basic message of the report is that high blood glucose suppresses expression of psoriasin. This is demonstrated elegantly using glucose clamped subjects, prediabetics, controls, and db/db mice. The authors show that, in turn, reduced psoriasin leads to increased bacterial growth in urine, and reduced bacterial clearance *in vivo*. The authors further show that high concentrations of glucose stimulate the translocation of YAP to the nucleus, a transcription factor involved in repression of psoriasin. In addition, elevated glucose concentrations induce the mannose receptor, which facilitates microbial adherence and represses the expression of occludin, which facilitates subepithelial microbial invasion. In addition, elevated glucose inhibited expression of IL1b and IL 6, both involved in expression of psoriasin. Curiously, estradiol stimulates psoriasin expression in the presence of high glucose concentration.

The manuscript is well written, and the data clearly presented. I found it useful that the authors included several responses to elevated glucose that predispose to bacterial infection, in addition to effects on psoriasin. Most important, from my perspective, is that adds to our understanding of the impact of high urinary concentrations of glucose on the risk of urinary tract infection.

I have no significant criticisms.

Answer: We thank the reviewer the time evaluating our manuscript and for appreciating the importance of our study with respect to the impact of urinary glucose and the risk of urinary tract infections.

Reviewer #2 (Remarks to the Author):

Nature Communications NCOMMS-21-38134

The present study describes the influence of hyperglycemia on psoriasin expression in the urinary bladder and whether it affects anti-infective response to *Escherichia coli*.

The authors analyze both patient's samples and mouse models to demonstrate that hyperglycemia decreases psoriasin expression and compromised the response to *E. coli*.

The study shows:

-In hyperglycemic patients and mice, psoriasin level is reduced in urine, serum, and bladder and this reduction was dependent on glucose and not insulin.

-Similar conclusions were obtained *in vitro* with uroepithelial cell line.

-High glucose decreases occludin expression in mouse bladder and in uroepithelial cell line and this decrease is rescued by addition of psoriasin.

-High glucose increases MRC1 and caveolin 1 expression after *E. coli* infection in mice and

uroepithelial cells favoring bacterial attachment.

-In uroepithelial cell line, high glucose decreases expression of RHOB a factor regulating actin assembly and increases nuclear YAP a factor downregulating psoriasin expression. Estradiol reverses this phenotype.

-High glucose decreased IL-6 expression via blocking IL-1 β expression, explaining the decreased expression of psoriasin known to be induced by IL-6.

-Diabetic mice showed impaired clearance of bacterial infection.

-Uroepithelial cell line infected with *E. coli* showed increased bacterial load after culture with high glucose and this phenotype is rescued by estradiol that increased psoriasin.

The present study may provide a mechanism explaining the observation that hyperglycemic patients have higher risk of recurrent bacterial infection. While interesting the study remains very puzzling and it is not obvious whether every data provided are connected to each other. Also, while one strength of the study is the use of both patient's samples and mouse model, the authors did not take advantage to the mouse model to clearly demonstrate the mechanism linking glucose to psoriasin and bacterial infection. Finally, the study is focused only on psoriasin and the redundancy with the other antimicrobial peptides (cathelicidin, mBD1) of which the expression is also decreased by glucose, is not considered.

Specific points:

Question: Figure 1: T1D and T2D patients should not be mixed since many other parameters additionally to hyperglycemia may influence psoriasin level in T1D patients.

Answer: We agree with the reviewer's concern that not only diabetes, but also other underlying factors might influence the psoriasin result. Therefore, clinical parameters like age, gender, BMI, HbA1c and duration of disease were considered separately for both T1D and T2D. The correlation between each of these parameters and the expression of psoriasin was analyzed, but no significant correlation was observed.

Plasma glucose levels may be equally increased in both T1D and T2D. Therefore, the results from both types of diabetes were merged since we focus on hyperglycemia as the most important common variable. However, we cannot rule out that in T2D there may be some degree of hyperinsulinemia. In T1D, serum insulin levels are also increased, due to insulin injections. However, as evident from our clamp study results, insulin does not seem to exert any impact on psoriasin (**Fig. 1d**). In summary, our combined findings strongly suggest that increased glucose levels, as found in both types of diabetes, is the major variable behind altered antimicrobial peptide expression and function. Based on this, we would like to keep the combined data of the T1D and T2D patients vs non-diabetic controls.

Question: Figure 1: In E-G, both groups (B6, db/db mice, uninfected, infected) should be shown together on the same graph. What is the significance of lower level of psoriasin observed in hyperglycemic condition in uninfected context?

Answer: It is well recognized that infection *per se* affects the expression of psoriasin. Therefore, we first wanted to show the effect of diabetes alone on psoriasin expression, followed by infection with *E. coli*. Lower levels of psoriasin in the uninfected state, resulted in lower occludin (**Fig. 3c, d**) and increased caveolin-1 (**Fig. 4c**) *in vivo*. We also demonstrated that lower psoriasin

levels *in vitro* resulted in compromised occludin (Fig. 3g, h), IL-1 β and IL-6 expression (Fig. 2d-f), with actin demonstrated rearrangement and YAP translocation (Fig. 5c-e). All these parameters were altered only by the effect of glucose and further contributed to an environment conducive with more *E. coli* growth. Therefore, we believe if we show the uninfected and infected graphs separately, the message will be clearer.

Question: Figure 1: What is the level of expression of other AMPs (cathelicidin, mBD1)?

Answer: We have analyzed the effect of glucose on other AMPs like *CAMP*, *DEFB1*, *DEFB4A*, *DEFB103A*, *RNASE7* (Fig. S1a). We only observed lower expression in *DEFB4A* and *RNASE7* along with *S100A7*. *DEFB4A* and *RNASE7* has been reported earlier and were therefore not of prime interest for us^{1,2}. *In vitro* we did not find any effect on *CAMP*, *DEFB1* and *DEFB103A*

Fig. S1a

Figure legend: Expression of different antimicrobial peptides in 6mM and 30mM glucose treated TERT-NHUC cells after 24 h.

We also analyzed the expression of *Cramp* and *Defb1* in urinary bladder of db/db mice and control mice, C57BL/6j, but we did not observe any significant difference between the groups. We have therefore not included the following data in this revised manuscript.

Figure legend: Expression of *Cramp* and *Defb1* in the urinary bladder of uninfected C57BL/6j and db/db mice.

Question: Figure 2: What is the mechanism of induction of occludin by psoriasin?

Answer: The Cys reduced form of psoriasin is a powerful endogenous zinc-chelator³. We therefore speculated that psoriasin could have a regulatory function in zinc homeostasis and its zinc-binding properties would determine induction or inhibition of the occludin gene expression⁴. To test the hypothesis that zinc-binding property of psoriasin is a possible candidate mechanism that could explain the observed effects, the cell penetrating and zinc specific chelator, N,N,N',N'-tetrakis (2-pyridylmethyl)ethylenediamine (TPEN) was used to treat human uroepithelial cells, TERT-NHUC, followed by treatment with high glucose for a total 24 h. TPEN treatment resulted in increased *OCLN* mRNA (**Fig. 3k**) similar to the effect of the psoriasin peptide (**Fig. 3j**) indicating a possible role of intracellular zinc depletion in the increased expression of *OCLN* in high glucose treated human uroepithelial cells.

Fig. 3k

Fig legend: Expression of *OCN* mRNA in TPEN and 30mM glucose treated TERT-NHUC cells.

Question: Decreased in occludin expression is directly mediated by glucose and/or via decreased of psoriasin?

Answer: Treatment of high glucose concentrations significantly decreased the occludin expression. To confirm the role of psoriasin in the regulation of occludin expression, we deleted *S100A7* coding for psoriasin using the crispr/cas9 system in human urothelial cells TERT-NHUC. To verify that *S100A7* was properly deleted and to rule out the influence of glucose on occludin, this experiment was performed in normal glucose treated cells. We observed that deletion of psoriasin resulted in a significant reduction in the expression of occludin even in normal glucose treated TERT-NHUC cells where both psoriasin and occludin express well, indicating the impact of psoriasin (**Fig. 3i**). The psoriasin influence was further supported in TERT-NHUC cells treated with high glucose when adding psoriasin peptide (**Fig. 3j**). We here conclude that psoriasin peptide treatment significantly increased *OCN* expression and confirm a strong effect of psoriasin even in high glucose.

Fig. 3i

Figure legend: Expression of psoriasin and occludin in *S100A7* deleted and control TERT-NHUC cells.

Further, psoriasin deletion did not induce any adverse effect on human uroepithelial cells, TERT-NHUC as evident from nuclear and actin cytoskeleton staining (**Fig. S2c**), confirming the above observed effect on occludin was primarily due to the loss of effect of psoriasin.

Fig. S2c

Figure legend: Phalloidin and DAPI staining in *S100A7* deleted and control TERT-NHUC cells.

Question: *Figure 3: The observation on MRC1 and caveolin-1 expression is interesting, however what is the link with psoriasin?*

Answer: The effect of the psoriasin peptide on the expression of MRC1 and caveolin-1 was analyzed in the revised manuscript. Interestingly, TERT-NHUC cells treated with high glucose and psoriasin peptide, demonstrated a significant decrease of *CAVI* (**Fig. 4e**), while no significant effect of psoriasin on *MRC1* was observed (**Fig. S2f**). In version 1 of this manuscript, we observed increased expression of Cav1 in high glucose treated TERT-NHUC cells (**Fig. 4d**) which resulted in increased bacterial uptake. In the revised version, we demonstrate that in the presence of psoriasin, a decrease of caveolin was observed (**Fig. 4e**), which might contribute to

significantly lower bacterial attachment of the pathogen onto the cell surface and a further decrease of bacterial uptake into the cell.

Fig. S2f and Fig. 4e

Figure legend: Expression of *MRC1* and *CAV1* mRNA in 30mM glucose treated TERT-NHUC cells, followed by 24 h psoriasin peptide treatment.

Question: Figure 3: What is the link between *RhoB* and YAP localization?

Answer: To confirm the link between *RhoB* and YAP localization, we transfected dominant negative *RHOB/T19N* and *RHOA/T19N* constructs as control separately in human urothelial cells, 5637. Significantly, more nuclear YAP/TAZ was observed in 30mM glucose as we have showed in version 1 of the manuscript (**Fig. 5c, e**). Interestingly, *RHOB/T19N* transfection resulted in even more nuclear YAP/TAZ, whereas no difference was observed in *RHOA/T19N* transfected cells (**Fig. 5f**). We here demonstrate a possible role of *RhoB* influencing the translocation of nuclear YAP/TAZ.

Fig. 5f

Figure legend: Expression of *RHOA/T19N* and *RHOB/T19N* in 30mM glucose treated 5637 cells, followed estimation of nuclear YAP/TAZ.

Question: Figure 3: The effect of estradiol is interesting however what is its mechanism of action?

Answer: We agree that the effect of estradiol in high glucose treated cells is very interesting and relevant in a clinical perspective. To identify the mechanism of action, we investigated the effect of estradiol on both IL-1 β and IL-6 regulation in hyperglycemic 5637 uroepithelial cells. We identified a significant upregulation of *IL6* mRNA (**Fig. 6j**), while no difference was observed for *IL1B* with estradiol treatment (**Fig. 6i**), indicating that estradiol acted on IL-6. Further, the effect of IL-6 on psoriasin revealed upregulation in high glucose treated uroepithelial cells and was demonstrated in version 1 of the manuscript (**Fig. 2m, n**).

Fig. 6i, j

Figure legend: Expression of *IL1B* and *IL6* mRNA in 30mM glucose treated 5637 cells, followed by 48 h of estradiol treatment

Question: Figure S4: Why these data that provide some mechanisms are in supplementary?

Answer: As per the suggestion by the reviewer, in the revised manuscript we have moved the old figure S4 from the Supplement to the main figure 2. We have also extended the figure with more mechanistic data to prove the effect of the IL-6 peptide in the regulation of psoriasin. IL-6 is known for its regulation of the pSTAT3 pathway. We observed that the levels of pSTAT3/STAT-3 was not altered with respect to high glucose and IL-6 treated TERT-NHUC cells (**Fig. 2g**). However, *SOCS3* (**Fig. 2h**) was compromised in high glucose treated TERT-NHUC cells. Therefore, we further investigated mechanisms that could downregulate SOCS3. Aryl hydrocarbon receptor (AhR) is known to downregulate SOCS3 in colonic epithelial cell⁵ and is upregulated by high glucose⁶ as reported in endothelial cells. Interestingly, we also observed higher *AHR* expression in human urothelial cells, 5637 treated with high glucose (**Fig. 2i**). Further, external supplementation of IL-6 restored the levels of *SOCS3* (**Fig. 2j**) in high glucose treated TERT-NHUC cells.

Fig. 2 g-j

Figure legend: Expression of pSTAT-3 and total STAT-3 using flowcytometry, *SOCS3* and *AHR* mRNA in 6mM and 30mM glucose treated TERT-NHUC and 5637 cells respectively, expression of *SOCS3* mRNA after 24 h of IL-6 peptide treatment in TERT-NHUC cells.

Question: *Figure S4: What would be the source of IL-1b in vivo? The uroepithelial cells?*

Answer: In the revised manuscript, we included the IHC of IL-1 β and IL-6 (**Fig. 2c**) *in vivo*. We observed a significantly lower expression of IL-1 β and IL-6 in superficial umbrella cells of db/db mice infected with *E. coli* which is in line with the result from urine exfoliated cells of diabetic patients (**Fig. 2a, b**). To conclude, superficial umbrella cells contribute with the majority of IL-1 β and IL-6 in the uroepithelium.

Fig. 2c

Figure legend: Immunohistochemistry of IL-1 β and IL-6 in *E. coli* infected urinary bladder of C57BL/6j and db/db at 24h.

Question: Figure S4: In vitro, does high glucose decreases IL-1b expression by uroepithelial cells and consequently IL-6 and psoriasin?

Answer: In vitro high glucose significantly downregulated the expression of *IL1B* mRNA (Fig. 2d). Blocking of IL-1 β using diacerein also resulted in lower expression of *IL6* (Fig. 2k) and *S100A7* (Fig. 2l) confirming IL-1 β and IL-6 mediated expression of psoriasin.

Fig. 2d, k-l.

Figure legend: Expression *IL1B* and *IL-6* mRNA in 6mM and 30mM glucose treated TERT-NHUC cells, and after 24 h of IL-1 β peptide and diacerein treated cells.

Question: Figure 4: These in vivo data should be presented earlier in the manuscript. However additional experiments should be performed: treatment of infected diabetic mice with psoriasin and estradiol.

Answer: In the present version of the manuscript, we showed the effect of high glucose on multiple parameters which resulted in increased rate of UTI. We further showed the effect of estradiol on bacterial infection as a possible therapeutic treatment strategy to prevent recurrent UTI. Therefore, we think Fig. 4 (now revised Fig. 6) is in the correct place as the concluding figure and we would like to keep its current place.

We agree with the reviewer that additional *in vivo* experiments with psoriasin and estradiol would be interesting and beneficial. However, these experiments merit detailed investigations which is beyond the scope of this manuscript.

Question: In vitro, whether psoriasin is more effective than other AMPs produced by uroepithelial cells to kill E. coli.

Answer: In all our experimental set ups and throughout the manuscript, we highlighted the effect of psoriasin only without comparing to any other antimicrobial peptides (AMPs). We claim that this is also a major AMP which is compromised in diabetes and resulted in increased rate of urinary tract infection. In line with this, previous literatures also suggest that among the AMPs produced by uroepithelial cells (psoriasin⁷, RNase7², hBD-1⁸), psoriasin⁹ is the most potent and abundant *E. coli*-cidal AMP. RNase 7 is 10-fold less potent than psoriasin against *E. coli*¹⁰. Its extremely high potency also against Gram-positive gut bacteria (*Enterococcus* ssp)¹⁰,

however, suggests a protective role against infection by *Ent. faecalis* and *Ent. faecium*. The cathelicidin, LL-37 is also 10-fold less potent than psoriasin against *E. coli* and additionally of lower abundance in unstimulated epithelial cells, suggesting that epithelial cathelicidin is less important relative to psoriasin to prevent *E. coli* infection. In urine, a major cellular source of cathelicidin are neutrophils. hBD-1 and mBD-1 are inactive as antimicrobials forms towards *E. coli* in its Cys-oxidized forms. Only its fully reduced (linearized) form can kill *E. coli* ¹¹.

Therefore, the relevance of psoriasin increases in the management of UTI and can effectively serve as a potential target for newer therapeutics in future.

Question: A graphical abstract should be very useful to summarize and link the different data described thorough the study.

Answer: We completely agree with the reviewer. In the revised manuscript, we have included a graphical summary in Fig. 7.

Figure legend: Impact of high glucose on uroepithelial cells, and in addition the effects of psoriasin, IL-1β, IL-6 and estradiol on the background of high glucose on uroepithelial cells.

Reviewer #3 (Remarks to the Author):

This manuscript by Mohanty *et al*, investigates how diabetes alters innate immune defenses

within the context of *E. coli* cystitis, a common infection in those with diabetes. The authors specifically focus on the antimicrobial peptide psoriasin (S100A7). They use 3 different models of study – samples from patients with or without diabetes (including those treated with hyperglycemic or euglycemic clamps), a mouse model of type 2 diabetes, and 2 uroepithelial cell lines. They found that high glucose concentrations induced lower psoriasin levels and impaired epithelial barrier through altered cytoskeletal function. These changes resulted in increased bacterial burdens including increased amounts of intracellular bacterial communities within bladder epithelial cells. Interestingly, exogenous replacement of psoriasin rescued some of these cytoskeletal abnormalities, and estradiol treatment led to enhanced levels of psoriasin. These findings will be of interest to those interested in host-pathogen interactions in the context of metabolic disease, which is becoming a very common risk for infections.

This paper is strong in several respects including its use of patient samples derived during the hyper/euglycemic clamps and use of multiple models to verify their findings. The methodology is appropriate to the studies conducted and the results and interpretation are clearly presented. This Reviewer’s enthusiasm for this otherwise strong manuscript is slightly lessened by a story that in places reads more as an unrelated list of immunologic abnormalities related to DM rather than a clear mechanistic change that drives these abnormalities.

Question: For instance, the connection between *Mrc1*, *RhoB*, caveolin-1 in relation to psoriasin is not made clear. Are the authors suggesting a series of cellular changes that culminate in psoriasin expression or a convergence of several different immunologic abnormalities that contribute to the phenotypes of infection?

Answer: As suggested by the reviewer, in the revised manuscript we have included the expression of *RHOB* (Fig. S3b), *MRC1* (Fig. S2f) and *CAVI* (Fig. 4e) in relation to psoriasin in high glucose treated cells. Psoriasin treatment resulted in downregulation of *CAVI* whereas no influence was observed on *RHOB* or *MRC1*. In the presence of psoriasin, changes in caveolin 1 might contribute to a significantly lower bacterial attachment of the pathogen onto the cell surface and further the bacterial entry into the cell.

Fig. S3b, S2f, 4e

Figure legend: Expression of *RHOB*, *MRC1* and *CAVI* mRNA in 30mM glucose treated TERT-NHUC cells, followed by 24 h psoriasin peptide treatment

The question raised by the reviewer is interesting and relevant. Our data suggest that the cellular changes primarily occurred due to high glucose, followed by compromised psoriasin expression. High glucose further downregulated occludin, however, psoriasin peptide treatment increased the occludin expression even if cells were hyperglycemic. We interpret our results that the psoriasin effect was stronger than the effect of glucose and able to outcompete the negative effects of glucose. Although a culminating effect can be observed, we believe that several different immunologic abnormalities can merge and thereby contribute to the increased rate of infection.

We have here only investigated the importance psoriasin in the diabetic condition, but there are reports of other antimicrobial peptides like RNase4¹², RNase7² and LCN2¹³ in the regulation of UTI in diabetes which further strengthens the convergence of several immunological observations in the final disease outcome.

Question: Are the functions of YAP and IL-6 inter-related or just alternative ways to regulate psoriasin expression?

Answer: We believe that the regulation of psoriasin by YAP and IL-6 are two possible alternative pathways. Interestingly, psoriasin induction is accompanied by YAP inactivation and are primarily dependent on the cell morphology and cell density. Psoriasin induction was primarily associated with phosphorylation in the YAP at the 127th serine residue (YAP-S127). It has also been reported that in suspended cells, increased phosphorylation in LATS1 will lead to psoriasin inhibition by the YAP pathway¹⁴.

However, in the IL-6 pathway analysis, we demonstrate that IL-1 β regulates IL-6, which in turn triggers psoriasin expression via AHR, pSTAT-3 and SOCS3 pathways (**Fig. 2**).

Further, to understand the inter-relation between IL-6 and YAP, in the revised manuscript, we analyzed the translocation of YAP after IL-6 peptide treatment in high glucose treated TERT-NHUC cells (**Fig. 5g**). Interestingly, we observed that IL-6 peptide treatment resulted in significantly less nuclear YAP when compared to high glucose treatment alone, justifying involvement of IL-6 and cytoplasmic phosphorylated YAP in the expression of psoriasin. Our combined results suggest that there are alternative pathways, but that they may still be interrelated.

Fig. 5g

Figure legend: Analysis of nuclear YAP/TAZ in IL-6 and high glucose treated TERT-NHUC cells.

Question: *The estradiol findings are intriguing, but it is unclear why this was tested, and it is not clear if estradiol provides broad protective effects via changes in glucose tolerance or triggers a specific pathway through psoriasin. The manuscript also felt like it ended abruptly without further investigating the in vivo effect of estradiol (or IL-6) on diabetic mice challenged with UTI.*

Answer: We agree with the reviewer that understanding the effect of estradiol on the glucose metabolism would deepen our understanding on possible ways to prevent recurrent UTIs in patients with diabetes. We have previously demonstrated the beneficial effect of estrogen on antimicrobial peptides and the effect on bacterial load during UTI¹⁵ in postmenopausal women and in mice as well as in *in vitro* studies. Because of the encouraging results, we wanted to investigate if similar effects were also seen in patients with diabetes. Hence, if estrogen could outcompete the negative effect of hyperglycemia. However, the effect of estrogens on glucose homeostasis is rather complex. It has been shown that estrogen receptors are important molecules involved in the adaptation of beta-cells to insulin resistance. This increases the demand on beta-cells to augment insulin secretion. Thus, estrogen receptor beta, potentiates glucose induced insulin secretion and thereby a suppression of serum glucose levels¹⁶.

To identify the mechanism of action, we investigated the effect of estradiol on both IL-1 β and IL-6 regulation in hyperglycemic 5637 uroepithelial cells. Although, there was no difference in *IL1B* mRNA (**Fig. 6i**), a significant upregulation of *IL6* mRNA (**Fig. 6j**) was observed with estradiol treatment, indicating that estradiol acted on IL-6. The effect of IL-6 on psoriasin further revealed upregulation in high glucose treated uroepithelial cells and was demonstrated in version 1 of the manuscript (**Fig. 2m, n**).

Fig. 6i, j

Figure legend: Expression of *IL1B* and *IL6* mRNA in 30mM glucose treated 5637 cells, followed by 48 h of estradiol treatment

To investigate the impact of IL-6 and estradiol *in vivo* would possibly shed light on this question and is relevant for future studies. Such experiments need to be performed in diabetic, estradiol treated, or diabetic and IL-6 deleted mice as well as corresponding control mice. Unfortunately, this is not possible due to logistic constraints. Although the effect of estrogen has been investigated, to our knowledge none of the studies have included the estradiol mediated effect on antimicrobial peptides in diabetes. Similarly, acute psychological stress has been shown to elicit IL-6 from brown adipocytes in mice. Then, IL-6 acts as a signal for causing hyperglycemia through hepatic gluconeogenesis¹⁷. This would further contribute to our understanding. However, we feel that this merit a detailed *in vivo* mechanistic analysis which is beyond the scope of this manuscript.

Major Critiques

Question: 1. How was the percent of cells with each type of actin filaments and quantification of nuclear vs cellular YAP determined? This does not appear to be described in the methods.

Answer: YAP nuclear localization analyzed by immunofluorescence microscopy after visualization with a mouse anti-YAP/TAZ antibody followed by an Alexa Fluor 488-conjugated anti-mouse antibody. The proportion of nuclear YAP/TAZ was calculated from microscopy images as follows: the integrated density module in ImageJ was used to measure the total fluorescence intensity of each cell. The same module was thereafter used to measure the fluorescence intensity in the nucleus of each cell. The proportion of nuclear YAP/TAZ over the total cellular YAP/TAZ was thereafter calculated.

Analysis of the type of actin filament organization was done by analyzing images acquired by immunofluorescence microscopy for the presence of short actin bundles, stress fibers or cortical actin as the dominant form of actin organization. Quantification was performed from three independent experiments per experimental condition. In the revised version, we have included this information.

Question: 2. The findings in figure 3 could use additional text to bridge connections between the cellular changes that authors identified and the consequences for the host-pathogen interaction they are examining. The authors seem to be suggesting that the upregulation of *Mrc1* and caveolin would increase adhesion and uptake by *E. coli* (potentially resulting in the IBCs seen in figure 4). What is the role of *RhoB* downregulation? Is this speculated to enhance IBC formation?

Answer: We first analyzed the effect of diabetes on *RHOB* expression, interestingly we observed a clear downregulation of *RHOB* mRNA in urine exfoliated cells of diabetes when compared to non-diabetic control (Fig. 5a). In previous studies, the role of decreased *RhoB* was associated with enhanced IBC formation in human bladder epithelial cells¹⁸. We therefore speculate that there is a direct link between the decreased *RhoB* and IBC observed. In the revised version, we have included the microscopic analysis of *RhoB* after 24 h infection in C57BL/6j and db/db mice bladder. *RhoB* expression was downregulated in diabetic mice bladders with significantly increased intracellular bacterial load (Fig. S3g).

Fig. 5a

Fig legend: Expression of *RHOB* mRNA in urine exfoliated cells of patients with diabetes (n=36) and non-diabetic control (n=20).

Fig. S3g

Fig legend: Expression of *RhoB* and *E. coli*, number of IBCs in bladders of C57BL/6j and db/db mice after 24 h infection.

Question: 2. Are the findings of RhoB downregulation connected to the actin filaments and YAP nuclear translocation?

Answer: To confirm the link between RhoB and YAP localization, we transfected dominant negative RHOB/T19N and RHOA/T19N constructs as control separately in human urothelial cells, 5637. Significantly, more nuclear YAP/TAZ was observed in 30mM glucose as we have already shown in version 1 of the manuscript (**Fig. 5c, e**). Interestingly, RHOB/T19N transfection resulted in even more nuclear YAP/TAZ, whereas no difference was observed in RHOA/T19N transfected cells (**Fig. 5f**). However, we did not observe any difference in the rearrangement of actin filaments after transfection of RHOA/T19N and RHOB/T19N (**Fig. S3f**). We here demonstrate a possible role of RhoB influencing the translocation of nuclear YAP/TAZ.

Fig. 5f

Figure legend: Expression of RHOA/T19N and RHOB/T19N in 30mM glucose treated 5637 cells, followed by estimation of nuclear YAP/TAZ.

Fig. S3f

Figure legend: F-actin staining after RHOA/T19N and RHOB/T19N transfection (marked with arrow) in 30mM glucose treated 5637 cells.

Question: In the discussion the authors speculate that high glucose concentrations triggers formation of more cortical actin which could potentiate the development of IBCs, but in figure 3G, estradiol did not improve the amount of cortical actin, but still resulted in less CFU in Figure 4K. How are the findings between relative CFU and actin structure related in this context?

Answer: To confirm the effect on CFU and actin structure, Latrunculin b was used to inhibit the actin nucleation. The actin rearrangement was inhibited before high glucose treatment which resulted in lower bacterial growth. This could be due to the dense actin network observed when compared to cells treated with 30mM glucose and is relevant as dense actin network prevents intracellular bacterial multiplication and IBCs formation¹⁹.

Although, estradiol altered the expression of actin with more cortical actin it did not result in increased CFU (Fig. S4f), primarily due to increased expression of the antimicrobial effect by psoriasin (Fig. 6k, l). The new result is now included in the revised manuscript.

Fig. S4f

Figure legend: *E. coli* survival assay performed in 30mM glucose pre-treated 5637 cells, with and without Latrunculin B (Lat B) and estradiol (E2) respectively, 2 h post infection.

Question: 3. Consider better summarizing if the authors think their findings represent a series of cellular changes that culminate in psoriasin expression or a convergence of several different immunologic abnormalities that contribute to the phenotypes of infection. A summary figure may be helpful in this regard.

Answer: We agree with the reviewer that a graphical summary is beneficial and has therefore been included as Fig. 7 describing the key findings that occurred due to high glucose in urothelial cells. Different immunological responses were also highlighted in presence of psoriasin, IL-1 β and IL-6 peptide and estradiol. We hope that this will make the understanding easier.

Figure legend: Impact of high glucose on uroepithelial cells, and in addition the effects of psoriasis, IL-1β, IL-6 and estradiol on the background of high glucose on uroepithelial cells.

Minor critiques

Question: 1. Data regarding *Mrc1* is actually present in Figure S2c and S2d (not S3c/d) as listed in the text (lines 160 and 163).

Answer: Thanks! In the revised manuscript, we have corrected the figure number.

Question: 2. Figure 3B needs to be larger, even on 150% magnification, the labels are barely readable. If needed, I would recommend removing the histogram and just showing the mean values as depicted in the insert.

Answer: As per the suggestion by the reviewer, we have enlarged the insert figures for better clarity and removed the histogram from all flow cytometry figures.

Question: 3. Figure 3F - the merged images of actin (red), dapi (blue), and YAP (green) IF staining are presented. The green staining (nuclear vs cellular) is difficult to discern due to the dapi staining, consider adding separate color channel images to supplementary figure 3.

Answer: We agree with the reviewer. For better clarity we have split the merged image and separately shown each protein and removed the DAPI staining. We now find that the figure has improved (**Fig. 5c** and **Fig. S3c**).

Fig. 5c

Fig. S3c

Figure legend: Expression of F-actin and YAP/TAZ in human urothelial cells 5637 and TERT-NHUC.

Question: 4. In Figure S2, for the db/db sample, there is no visible red staining, though the quantification suggests the occludin staining should be similar to that seen in Figure 2D or F? Does this image include the red channel?

Answer: In the revised manuscript, we have replaced the old figure and added an alternate figure with visible red staining.

Fig. S2b

Figure legend: Expression of occludin in the urinary bladder of C57BL/6j and db/db, 7 days post *E. coli* infection.

Reviewer #4 (Remarks to the Author):

Summary and General Comments: This paper by Mohanty *et al* concerns the effects of glucose on the anti-microbial peptide psoriasin and *E. coli* burden in the urinary bladder. They find that high glucose and not insulin reduces the psoriasin levels and impairs epithelial barrier function with altered cell membrane proteins and cytoskeletal organization. This was associated with increased bacterial burden reflected by increased intracellular bacterial communities. They show the high glucose impairs the release of proinflammatory cytokine IL-6 and IL1B which they postulate affects the psoriasin levels. Estrogen treatment restored the effects of hyperglycemia. They conclude that the high glucose seen in diabetes underlies compromised innate immunity in the bladder.

This is an interesting study that is well focused on innate immune mechanisms in the bladder. The authors suggest that the findings may explain the relationships between high glucose and impaired immunity that leads to urinary tract infections. There are a few specific questions about the data presentation, described below, but overall, it is well presented. The conclusions that are drawn are supported by the presented data. A major limitation of the work is the absence of a clear mechanism whereby glucose affects the innate mediators.

Question: They suggest that IL-6 may be an important intermediate but the way in which IL-6 does this (through STAT3??) and how glucose affects IL-6 levels are not studied.

Answer: We have now extended the current figure 2 (previous version figure S4) with more mechanistic data to prove the effect of the IL-6 peptide in the regulation of psoriasin. IL-6 is known for its regulation of the pSTAT3 pathway. We observed that the levels of pSTAT3/STAT-3 was not altered with respect to high glucose and IL-6 treated TERT-NHUC cells (**Fig. 2g**). However, *SOCS3* (**Fig. 2h**) was compromised in high glucose treated TERT-NHUC cells. Therefore, we further investigated mechanisms that could downregulate *SOCS3*. Aryl hydrocarbon receptor (AhR) is known to downregulate *SOCS3* in colonic epithelial cell⁵ and is upregulated by high glucose⁶ as reported in endothelial cells. Interestingly, we also observed higher *AHR* expression in human urothelial cells, 5637, treated with high glucose (**Fig. 2i**). Further, external supplementation of IL-6 restored the levels of *SOCS3* (**Fig. 2j**) in high glucose treated TERT-NHUC cells.

Fig. 2g-j.

Figure legend: Expression of pSTAT-3 and total STAT-3 using flow cytometry, *SOCS3* and *AHR* mRNA in 6mM and 30mM glucose treated TERT-NHUC and 5637 cells respectively, expression of *SOCS3* mRNA after 24 h of IL-6 peptide treatment in TERT-NHUC cells.

In vitro high glucose significantly downregulated the expression of *IL1B* (Fig. 2d). Blocking of IL-1 β using diacerein in high glucose conditions also resulted in lower expression of *IL6* (Fig. 2k) and *S100A7* (Fig. 2l) confirming IL-1 β and IL-6 mediated expression of psoriasis.

Fig. 2d, k-l.

Figure legend: Expression *IL1B* and *IL-6* mRNA in 6mM and 30mM glucose treated TERT-NHUC cells, and after 24 h of IL-1 β peptide and diacerein treated cells.

Question: The significance of the findings and relevance to disease is not clear. Based on their proposition, there should be a relationship between glucose control (measured by HbA1c levels) and psoriasis or *S100A7* levels in the urine and urine cells as shown in Figures 1a, b. It would seem that this question should at least be addressed since the investigators have the data (Suppl Table 1).

Answer: We analyzed the HbA1c levels with respect to the expression of psoriasin in the urine exfoliated cells and urine. However, we did not observe any correlation. Similarly, other clinical parameters like age, gender, BMI, and duration of disease were considered separately for all included patients. Correlation of these parameters were measured with the level of expression of psoriasin, but no significant correlation was observed. HbA1c levels are reflecting blood glucose control during the latest 4-6 weeks. Despite no correlation between HbA1c and psoriasin levels, it is very likely that occasionally elevated blood and urine glucose levels, especially post-prandially, may impact the psoriasin expression.

In T1D, serum insulin levels are also increased, due to insulin injections. However, as evident from our clamp study results, insulin does not seem to exert any impact on the psoriasin expression. In summary, our combined findings strongly suggest that increased glucose levels, as found in both types of diabetes, is the major variable behind altered antimicrobial peptide expression and function.

Question: *In addition, they perform a euglycemic hyperinsulinemic clamp in patients with T1D and do not find a difference in the psoriasin levels and claim that insulin does not affect the psoriasin levels. However, the studies could have more precisely addressed this question by a euglycemic or hyperglycemic clamp with the same insulin delivery in patients with T1D who do not make their own insulin. Likewise, clamp studies could be done in rodents to answer the question about the relationship between insulin and psoriasin.*

Answer: A hyperglycemic clamp is designed to investigate insulin secretion in response to glucose. Such a clamp is not useful in T1D patients who lack insulin secretory capacity. To study the effect of insulin, we believe that we have used an optimal experimental design, i.e. normoglycemic hyperinsulinemic clamps, in which the mean steady state insulin levels is fifty fold higher²⁰. This allowed us to draw the conclusion that the insulin concentration did not influence psoriasin expression. We also believe that our results in human adults are more valid and stronger in this context than those possible to obtain from animal studies.

Question: *More challenging to understand is the timing of the changes in the psoriasin levels and changes in glucose. They describe changes that occur within minutes. It leaves the reader to wonder if the changes they describe (Figure 1i) are of significance. Finally, the restoration of the effects of hyperglycemia with estrogen also raise question about the clinical significance. Women with diabetes are more than twice as likely at non-diabetic women to have a urinary tract infection and these infections are more common in women than men.*

Answer: Glucose treatment significantly downregulated the expression of psoriasin at all time points tested following 24 h of glucose treatment. Only after *E. coli* infection, the expression changes within minutes, which was even more pronounced in cells treated with normal glucose concentrations in all tested time points. After an initial rapid peak, the reduction at later time points could therefore be attributed either to inhibition of psoriasin transcription or to mRNA decay exceeding synthesis in normoglycemic cells.

However, high glucose treated cells did not respond well to the *E. coli* infection, confirming a compromised state of the cells. Previously we have also reported expression of cathelicidin within 5 minutes of *E. coli* infection in uroepithelial cells, UROtsa²¹.

Restoration of psoriasin expression with estradiol was confirmed after 48h of pre-treatment. Therefore, we think that vaginal estrogen substitution will be possible to use to achieve

restoration of psoriasin expression in diabetic females, especially in those having recurrent UTI. We completely agree with the reviewer that sex-based differences largely contribute to the occurrence of UTI. Although frequency of UTI in females are high, UTI has been shown to be more persistent in male. Interestingly, female mice had a more robust innate immune response with higher IL-17 expression²². Moreover, it is also known that estrogen receptor signaling affects psoriasin expression²³. Therefore, we believe in a strong clinical impact of estradiol in the regulation of a robust immune response.

Specific points:

Question: 1. A common statistical approach that is used is to look at fold change with an experimental condition compared to a baseline. Presenting the data in this way (eg. Fig 2i, Fig S5e) does not allow us to see the variance in the data at the baseline. Furthermore, some values increase and some decrease and the relationship between the baseline value and the change cannot be appreciated in the way this data are presented. It appears that what they actually did is a 1 sample t-test comparing the fold change to 1. The statistical approach should be clarified.

Answer: In the revised manuscript, we have included the statistical analysis performed for each figure in the figure legends. For some of the figures, like adhesion and survival (for e.g **fig S4e**) and immunohistochemistry (for e.g **fig 1f**) we have normalized the control to 1, in order to highlight the relative difference between high glucose and normal glucose treated cells. However, in other experiments like mRNA analysis in urine exfoliated cells (**Fig. 1a**) or bladder mRNA (for e.g **Fig. 1e**) we have presented individual data as the Reviewer suggested. Our statistical methods were verified after consultation with the Karolinska Institute statistician.

Question: 2. Fig1B: How did body weight affect the psoriasin concentrations? What is the appropriate control group?

Answer: Bodyweight as analyzed by BMI did not influence the psoriasin concentration and there was no correlation. In fig 1b, we showed the urine psoriasin levels of samples from both diabetic and non-diabetic persons, and the relative comparison was performed with respect to urine creatinine levels.

Question: 3. Fig 1C – the changes in the plasma psoriasin are modest despite the statistical significance.

Answer: We agree with the reviewer that the changes were modest although statistically significant. The sample size is limited which impact the results, but out of 19 samples analyzed, 17 samples showed downregulation with glucose treatment. Paired analysis confirmed a significant difference.

Question: 4. In Fig S2, db/db mice are compared to B6 mice (which are all assigned 1). However, these are two different strains so a paired test here is not appropriate.

Answer: We agree with the reviewer. As suggested, in the revised manuscript for all immunohistochemistry performed in mouse, results have been replaced with unpaired non-parametric t test. Appropriate tests performed are mentioned in the figure legends.

Question: 5. Throughout the paper relative mRNA levels are presented. However, in the methods, RT-PCR is described (line 456). How was mRNA measured? If it was RT-PCR, what was the reference cDNA and what data are presented.

Answer: For RT-PCR analysis, total RNA was isolated and measured using nanodrop (Thermo). 0.1 to 500 ng of RNA was used for the preparation of cDNA using standard cDNA synthesis kit. The cDNA was used for expression analysis, relative expressions were compared to control cells using reference housekeeping genes. A detailed description of the method is included in the revised draft.

Question: 6. Fig S1 – psoriasin is not different in DM vs non-diabetic controls??? Where is this measured?

Answer: In fig. S1 the analysis was performed in serum, where we observed a trend of lower expression in patients with DM compared to control persons, which however, did not reach significance. On the contrary, our results from urine exfoliated cells and urine samples (**Fig. 1a, b**) were clearly and significantly downregulated. The lack of significance in serum might be due to the very strict control of patients with diabetes as evidenced by the glycated hemoglobin A1c (HbA1c) values (**Suppl. table 1**).

Question: 7. In Fig S5e: We don't know the baseline levels (at 6mM). If fold levels are to be compared, why are the 6 mM data presented or if they are presented, we should see the actual values.

Answer: We have normalized the control to 1 to highlight the relative difference between high glucose and normal glucose treated cells.

References

1. Brauner H, *et al.* type 2 diabetes mellitus and the effect of the anti-oxidant coenzyme Q10 on inflamMarkers of innate immune activity in patients with type 1 and matory activity. *Clin Exp Immunol* **177**, 478-482 (2014).
2. Eichler TE, *et al.* Insulin and the phosphatidylinositol 3-kinase signaling pathway regulate Ribonuclease 7 expression in the human urinary tract. *Kidney Int* **90**, 568-579 (2016).
3. Hein KZ, *et al.* Disulphide-reduced psoriasin is a human apoptosis-inducing broad-spectrum fungicide. *Proceedings of the National Academy of Sciences of the United States of America* **112**, 13039-13044 (2015).

4. Miyoshi Y, Tanabe S, Suzuki T. Cellular zinc is required for intestinal epithelial barrier maintenance via the regulation of claudin-3 and occludin expression. *American journal of physiology Gastrointestinal and liver physiology* **311**, G105-116 (2016).
5. Han H, *et al.* Loss of aryl hydrocarbon receptor suppresses the response of colonic epithelial cells to IL22 signaling by upregulating SOCS3. *American journal of physiology Gastrointestinal and liver physiology* **322**, G93-g106 (2022).
6. Dabir P, Marinic TE, Krukovets I, Stenina OI. Aryl hydrocarbon receptor is activated by glucose and regulates the thrombospondin-1 gene promoter in endothelial cells. *Circulation research* **102**, 1558-1565 (2008).
7. Ostergaard M, Wolf H, Orntoft TF, Celis JE. Psoriasin (S100A7): a putative urinary marker for the follow-up of patients with bladder squamous cell carcinomas. *Electrophoresis* **20**, 349-354 (1999).
8. Becknell B, *et al.* Expression and antimicrobial function of beta-defensin 1 in the lower urinary tract. *PloS one* **8**, e77714 (2013).
9. Glaser R, Harder J, Lange H, Bartels J, Christophers E, Schroder JM. Antimicrobial psoriasin (S100A7) protects human skin from Escherichia coli infection. *Nat Immunol* **6**, 57-64 (2005).
10. Harder J, Schroder JM. RNase 7, a novel innate immune defense antimicrobial protein of healthy human skin. *The Journal of biological chemistry* **277**, 46779-46784 (2002).
11. Schroeder BO, *et al.* Reduction of disulphide bonds unmasks potent antimicrobial activity of human β -defensin 1. *Nature* **469**, 419-423 (2011).
12. Bender K, *et al.* Expression and function of human ribonuclease 4 in the kidney and urinary tract. *American journal of physiology Renal physiology* **320**, F972-f983 (2021).
13. Murtha MJ, *et al.* Insulin receptor signaling regulates renal collecting duct and intercalated cell antibacterial defenses. *The Journal of clinical investigation* **128**, 5634-5646 (2018).
14. Li Y, *et al.* S100A7 induction is repressed by YAP via the Hippo pathway in A431 cells. *Oncotarget* **7**, 38133-38142 (2016).
15. Luthje P, *et al.* Estrogen supports urothelial defense mechanisms. *Sci Transl Med* **5**, 190ra180 (2013).
16. Nadal A, *et al.* Role of estrogen receptors alpha, beta and GPER1/GPR30 in pancreatic beta-cells. *Frontiers in bioscience (Landmark edition)* **16**, 251-260 (2011).
17. Qing H, *et al.* Origin and Function of Stress-Induced IL-6 in Murine Models. *Cell* **182**, 1660 (2020).

18. Moorthy S, Byfield FJ, Janmey PA, Klein EA. Matrix stiffness regulates endosomal escape of uropathogenic E. coli. *Cell Microbiol* **22**, e13196 (2020).
19. Eto DS, Sundsbak JL, Mulvey MA. Actin-gated intracellular growth and resurgence of uropathogenic Escherichia coli. *Cell Microbiol* **8**, 704-717 (2006).
20. Donga E, Dekkers OM, Corssmit EP, Romijn JA. Insulin resistance in patients with type 1 diabetes assessed by glucose clamp studies: systematic review and meta-analysis. *European journal of endocrinology* **173**, 101-109 (2015).
21. Chromek M, *et al.* The antimicrobial peptide cathelicidin protects the urinary tract against invasive bacterial infection. *Nat Med* **12**, 636-641 (2006).
22. Zychlinsky Scharff A, *et al.* Sex differences in IL-17 contribute to chronicity in male versus female urinary tract infection. *JCI insight* **5**, (2019).
23. Skliris GP, *et al.* Estrogen receptor-beta regulates psoriasin (S100A7) in human breast cancer. *Breast cancer research and treatment* **104**, 75-85 (2007).

Reviewers' Comments:

Reviewer #2:

Remarks to the Author:

The authors have addressed all my concerns I raised in my previous review.

Reviewer #3:

Remarks to the Author:

This revised manuscript by Mohanty et al. has addressed this Reviewer's critiques from the original submission. This Reviewer thanks the authors for their detailed responses. The additional summary figure provides a nice closing to the detailed findings of this manuscript.

Reviewer #4:

Remarks to the Author:

The authors have done considerable work in addressing the Reviewers' concerns. I do not have further concerns.

Responses to Reviewers.

Reviewer #2 (Remarks to the Author):

The authors have addressed all my concerns I raised in my previous review.

Answer: We thank the reviewer for the insightful feedback and for taking the time reviewing the manuscript.

Reviewer #3 (Remarks to the Author):

This revised manuscript by Mohanty et al. has addressed this Reviewer's critiques from the original submission. This Reviewer thanks the authors for their detailed responses. The additional summary figure provides a nice closing to the detailed findings of this manuscript.

Answer: We thank the reviewer for the time invested and the relevant suggestions which have significantly improved the manuscript.

Reviewer #4 (Remarks to the Author):

The authors have done considerable work in addressing the Reviewers' concerns. I do not have further concerns.

Answer: Thank you for the constructive suggestions, appreciation of our work and for the time spent on reviewing.